# A Model to Calculate Fatigue Damage Caused by Partial Waking during Wind Farm Optimization

Andrew P. J. Stanley[1], Jennifer King[2], Christopher Bay[2], and Andrew Ning[3]

[1](a) currently at: National Renewable Energy Laboratory, National Wind Technology Center, Boulder, CO 80303 USA
(b) formerly at: Department of Mechanical Engineering, Brigham Young University, Provo, UT 84602 USA
[2]National Renewable Energy Laboratory, National Wind Technology Center, Boulder, CO 80303 USA
[3]Department of Mechanical Engineering, Brigham Young University, Provo, UT 84602 USA

**Correspondence:** Andrew P. J. Stanley (PJ.Stanley@nrel.gov)

**Abstract.** Wind turbines in wind farms often operate in waked or partially waked conditions, which can greatly increase the fatigue damage. Some fatigue considerations may be included, but currently a full fidelity analysis of the increased damage a turbine experiences in a wind farm is not considered in wind farm layout optimization because existing models are too computationally expensive. In this paper, we present a model to calculate fatigue damage caused by partial waking on a wind turbine that is computationally efficient and can be included in wind farm layout optimization. The model relies on analytic velocity, turbulence, and loads models commonly used in farm research and design, and captures some of the effects of turbulence on the fatigue loading. Compared to high-fidelity simulation data, our model accurately predicts the damage trends of various waking conditions. We also perform example wind farm layout optimizations with our presented model in which we maximize the annual energy production (AEP) of a wind farm while constraining the damage of the turbines in the farm. The results of our optimization show that the turbine damage can be significantly reduced, more than 10%, with only a small sacrifice of around 0.07% to the AEP, or the damage can be reduced by 20% with an AEP sacrifice of 0.6%.

## 1 Introduction

Modern wind turbines are some of the largest machines in the world. Improvements in materials and technologies in recent years have allowed for taller towers, longer blades, and more power output (Wiser et al., 2016; Enevoldsen and Xydis, 2019). Because of their large size and associated large loads, as well as the cyclic loading caused by their rotational operation, fatigue

is a vital consideration in wind turbine design (Hübler et al., 2019). Turbines must be designed to operate without failure and with minimal maintenance for the duration of their lifetime, which is usually 20–25 years (Hu et al., 2016; Ziegler et al., 2018). The cyclic load variations experienced by a wind turbine can be exacerbated when many turbines are built relatively

close together in a wind farm. Wind turbines extract momentum from the moving air, creating a wake of slow-moving wind behind them. In wind farms, turbine wakes can cause an uneven distribution of wind speeds across the swept rotor areas of downstream turbines, which intensifies the load fluctuations already present from turbulence, gravity, and wind shear. To make matters worse from a structural loads perspective, wind farms are usually optimized for maximum power production. Wind farm optimization can be used to refer to turbine layout optimization when constructing the farm or active yaw control to steer

wakes away from downstream turbines. In each case, the objective is typically to maximize power by reducing the velocity deficits caused by wakes. This optimization often leads to partially waked turbines, which can be desirable for increasing power but devastating for the structure, causing turbines to fail earlier than expected and increasing overall costs. To account for increased fatigue damage caused by partial waking in wind farms, we developed a reduced-order model to quickly calculate loads, which can be used to constrain turbine damage in an optimization framework.

Because wind turbines are large investments and their design is driven by fatigue, many researchers have studied how different conditions affect wind turbine loading and fatigue. An early study by Thomsen and Sørensen used field data and the aeroelastic code HawC to examine how different atmospheric and waking conditions affect wind turbines. They found that fatigue loading increases by 5%–15% when a turbine is operating in a wake compared to when it is operating in the freestream (Thomsen and Sørensen, 1999). A more recent paper by Meng et al. finds similar results to the study performed by Thomsen

and Sørensen. They used the large eddy simulation code Simulator fOr Wind Farm Applications (SOWFA) and the finite element analysis code BECAS to find that, for an open-source reference turbine, the fatigue loads increased by 16% when the turbine operated in a wake compared to the freestream (Meng et al., 2019). An additional study by Kim et al. found that when the turbulence intensity was increased from around 12% to around 20% in a wind farm, the fatigue loads increased between 30%–50% (Kim et al., 2015). This study highlights the importance of turbulence in calculating the fatigue on a wind turbine.

All three of these studies indicate that loading and fatigue are greatly affected in wind farms where waking and partial waking are normal operating conditions for many turbines in the farm.

In addition to characterizing fatigue loading in different conditions, several studies used active control strategies to reduce fatigue loading on wind turbines. In one of the first studies on using turbine control to reduce loads, Bossanyi showed that individual blade pitch control can be used to significantly reduce loading on the turbine structure (Bossanyi, 2003). Njiri et

al. developed a control method in which the power production is slightly sacrificed to alleviate loads on the turbine structure. Near the end of a wind turbine's usual lifetime, the generator can be derated to extend its lifetime. The bending moments on the blades can be reduced by more than 35% by derating the generator from 100% to 70% (Njiri et al., 2019). Bernhammer et al. performed a study with smart rotors, or rotors that use active aerodynamic devices (like flaps), to alter flow. They found that by using smart rotors, the loads can be reduced by 5%–15% (Bernhammer et al., 2016).

The studies mentioned above use active control of wind turbines to reduce the loads experienced by wind turbines, with the implicit assumption that the inflow to the wind turbine cannot be controlled. However, the inflow can be changed if considered

during the wind farm layout design phase or through wake steering. Mendez Reyes et al. presented a study in which a look-up table was used to quickly compute turbine loads in wind plants that use active turbine control. The approach used high fidelity, computationally expensive methods to calculate a range of turbine loads, then used interpolation and a look-up table to quickly calculate the turbine loads during turbine control optimization (Mendez Reyes et al., 2019). Like this study indicates, with appropriate models, the loads experienced by each wind turbine in a farm can be predicted and constrained during optimization. Current fatigue load prediction models are computationally expensive and not suitable for use in an optimization framework or, in the case of the study by Mendez Reyes et al., requires a large amount of computationally expensive loads calculations before the optimization can be run. In this paper, we present a model to quickly calculate load histories on wind turbine blades and the associated fatigue damage. The presented model is fast enough to be used in a wind farm layout optimization and predicts the damage trends for different waking and partial-waking conditions well compared to higher-fidelity methods. Additionally, we demonstrate the application of our newly presented model in example wind farm layout optimizations and show how including fatigue damage constraints changes the results of the optimization.

## 2   Wind Turbine Loads Model

In this section, we describe the concepts and methods we used to estimate the loads and fatigue damage at the blade root of a wind turbine and how the various steps and models fit together. For the models and results shown in this paper, we used the NREL 5-MW reference turbine, which is an open-source turbine design used in many research studies (Jonkman et al., 2009). Note that we have validated the various parts of our model with the large eddy simulation software, SOWFA (Churchfield and Lee, 2012), and the aeroelastic structural analysis software OpenFAST (National Renewable Energy Laboratory, 2017). Both of these programs are open source and created at the National Renewable Energy Laboratory (NREL). We used the velocity data from SOWFA as the velocity input into OpenFAST to calculate the load histories on a wind turbine for various waking conditions and compared the associated damage with these load histories to our proposed model. SOWFA has been validated against real wind farm flow data, and has been shown to accurately represent wind speed, direction, and turbulence in a farm (Churchfield et al., 2012a, b). Refer to Appendix A for more details about our SOWFA simulations.

Because we have compared all of our intermediate models, as well as the final damage calculations, to the high-fidelity SOWFA and OpenFAST data, one might wonder why we did not directly use some surrogate of the SOWFA and OpenFAST data instead of the lower-fidelity intermediate models, or even create a surrogate directly of the final fatigue damage. These possible methods would likely provide accurate results, and a surrogate would be computationally efficient for use during an optimization. However, a primary purpose of our model is to provide a method to estimate fatigue damage while leaving open the possibility of using computationally efficient analytic models. Our model does not require the user to run computationally expensive, complex, and high-fidelity simulations, although they certainly could. With our method, simple analytic models can be used to sufficiently estimate fatigue damage from partial waking, given that the intermediate analytic models are sufficiently accurate. For this paper, we use tuning constants to improve the comparison of our analytic models to our SOWFA data, which

did require us to generate the high-fidelity data. However, for many or most applications where this fatigue model, such model
tuning and exact match of previously generated data would be unnecessary.

The rest of this section will discuss the various details of the loads and damage model that we present in this paper. Fatigue
damage on a wind turbine is caused by cyclic loading as a turbine rotates. These load variations are caused by gravity, uneven
wind speeds across the rotor caused by partial waking, and turbulence. In fact, the loads on a turbine are more complicated
than this because they depend on the interactions of all of these causes. For example, a partially waked turbine also subjected to
high turbulence will experience more extreme load fluctuations than one with a uniform inflow. To account for the interactions
of all of these fatigue drivers, our model predicts the loads on a turbine blade at a predetermined number of azimuth angles
and blade rotations. The predicted load history is then used to calculate the fatigue damage a turbine blade experiences for a
given turbine layout and wind condition. Figure 1 shows a general overview of the model. Because each part of the model is
important and has some subtleties, each will be discussed individually.

The notation used throughout this paper is described alongside the first use of each variable, which in general should be easy
to follow. However, there are a few specific cases that we will also mention here. There are several different wind speeds that
need to be differentiated. A capital $U$ refers to the wind speeds calculated with an analytic wake model, which can be thought
of as a time averaged wind speed value. The lower-case $u$ is an instantaneous wind speed, which accounts for turbulence.
There are several specific wind speeds that are mentioned throughout this paper, though we feel 3 specifically are important to
differentiate and explain: $U$ alone is the wind speed at a single point, $U_{\mathrm{rotor}}$ is the average wind speed across the entire rotor, and
is referred to as the average rotor wind speed or the turbine inflow wind speed, and $U_{\mathrm{blade}}$ is the average wind speed acting on
a turbine blade. In addition to the wind speeds, the turbulence intensity is also differentiated between point values, TI, values
affecting the entire rotor, $\mathrm{TI}_{\mathrm{rotor}}$, and values affecting a single blade, $\mathrm{TI}_{\mathrm{blade}}$.

## 2.1 Generate Turbulence Samples

One large driver of fatigue damage in a wind turbine is turbulence. The severity of turbulence is usually described with
turbulence intensity, which is a measure of the standard deviation of the wind speed divided by its mean. This provides a
time-averaged description of the turbulence, but provides no information about the instantaneous values. In order to account
for the instantaneous values, we created a set of turbulence samples, $S$, which is some set of values with a standard deviation
of one and mean value of zero. These samples are scaled later by the local turbulence intensity and the mean wind speed to
obtain instantaneous velocities and loads.

There are a variety of possible methods that could be used to define these turbulence samples. For the results shown in this
paper, we assumed that the velocity variations due to turbulence were Gaussian, and used Latin hypercube sampling to sample
from a normal distribution with a mean of zero and a standard deviation of one. We then shuffled the samples such that the
difference of sequential values in the distribution was also approximately Gaussian. This was important because the fatigue
damage depends on the order in which loads appear. The turbulence samples used for this paper are shown in Fig. 2.

Although the turbulence samples used in this paper have the important statistical qualities required for the fatigue calculations
in this paper, there are a variety of other methods that could be used to generate the turbulence values. One method could be

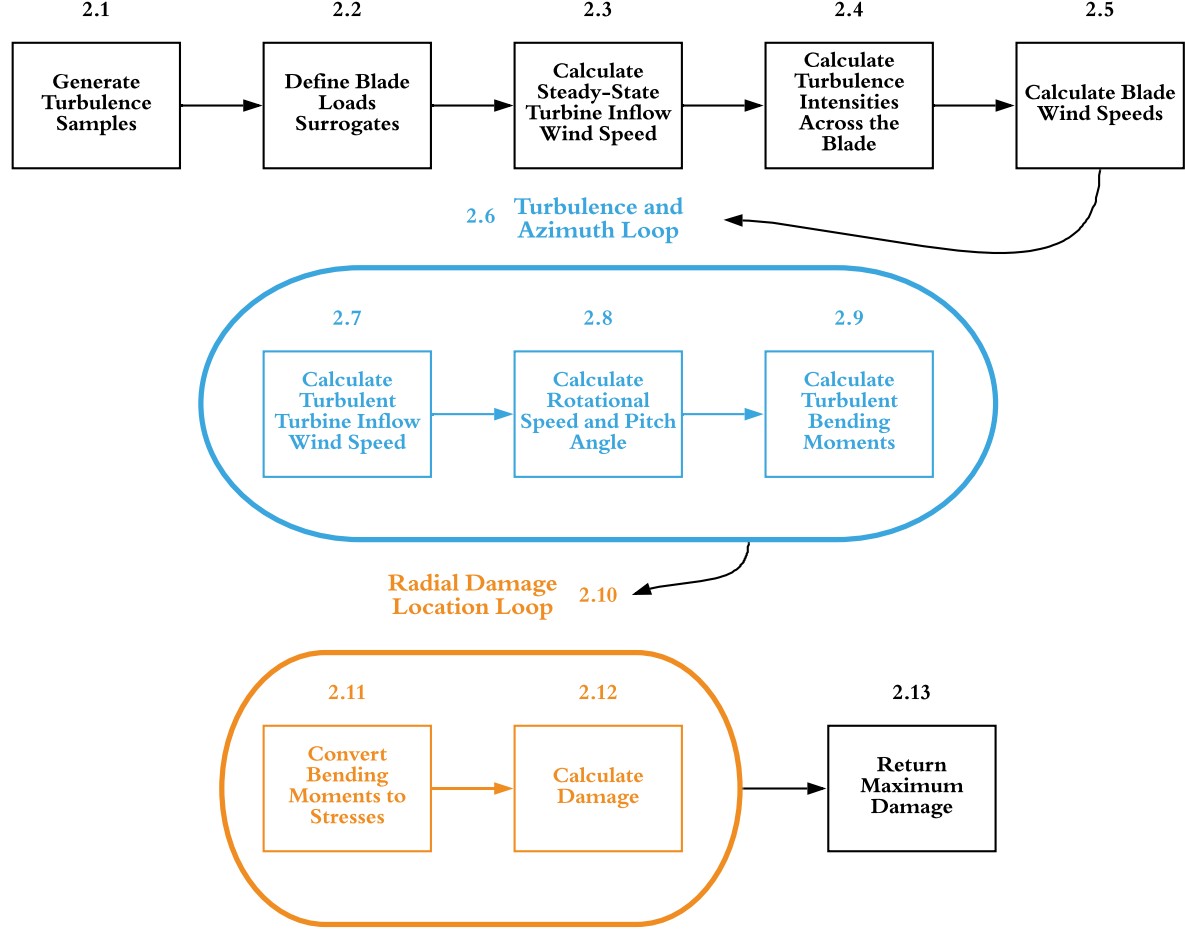

**Figure 1.** A flow chart of the damage calculation model used in this study.

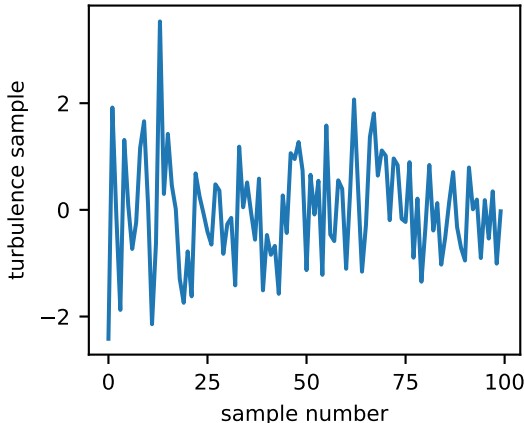

**Figure 2.** The set of turbulence samples, $S$, used in this study. Turbulence intensity is defined as $\text{TI} = \sigma_u/\bar{u}$, where $\sigma_u$ is the standard deviation in wind speeds over a given time, and $\bar{u}$ is the mean wind speed. These turbulence samples are used in a future step to calculate an instantaneous wind speed adjusted for turbulence as $u_i = U_{\text{steady}}(1 + S_i\,\text{TI})$.

to use the Sandia method, also known as the Veers method, introduced in 1988 Veers (1988). Another could be to us the turbulence generator TurbSim to generate the turbulence samples, which has made several improvements since the Sandia
method was introduced (Jonkman and Buhl Jr, 2006). Using one of these methods could create more realistic turbulence history, but requires using an external program. For the results shown in this paper, the turbulence samples we generated are sufficient for demonstrating our method, and had appropriate statistical properties to compare well with high fidelity simulations.

### 2.2 Define Blade Loads Surrogates

The next step is to create a surrogate that defines how the bending moments on the blade scale with the undisturbed wind speed
and blade pitch angle. In order to account for the effect of turbulence in the load history on a turbine blade, we needed to calculate the loads at many different wind speeds for a certain turbine blade orientation. Directly calculating these loads may be time consuming and not appropriate for an optimization framework where the model may need to be called hundreds or thousands of times. Thus, we approximated the blade loads with piecewise surrogate functions that are a function of the wind speed acting across the blade, and the pitch angle. To accomplish this, we used CCBlade, a blade element momentum method
for propellers and turbines (Ning, 2020). A higher fidelity model could also be used to calculate the loads for this step, and our choice to use CCBlade was to allow for an easy transition to evaluating the loads directly in the optimization loop if desired. We calculated the loads on the blades for various wind speeds and pitch angles in order to approximate representative functions. We needed a loads surrogate for both the flatwise and edgewise loads for each of the azimuth angles that were considered in the damage model (these azimuth angles are further discussed in Sec. 2.6).

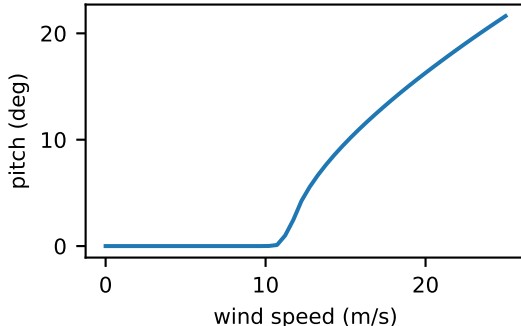

**Figure 3.** Blade pitch angle versus wind speed for the NREL 5-MW reference turbine.

The loads on a turbine blade are dependent on the wind speeds, the rotation speed, and the blade pitch angle. When creating the surrogate models, we assumed the wind speed was constant across the blade for each speed that we evaluated. The rotation speed was determined from the ideal tip-speed ratio of the wind turbine:

$$\Omega = \min\left(\lambda U_{\text{rotor}}/R, \ \Omega_{\text{max}}\right) \tag{1}$$

where $\Omega$ is the rotation speed; $\lambda$ is the tip-speed ratio from the turbine definition; $U_{\text{rotor}}$ is the inflow wind speed to the rotor; $R$
is the rotor radius; and $\Omega_{\text{max}}$ is the maximum turbine rotation speed (7.55, 63.2 meters, and 12.1 rpm for the 5-MW reference turbine, respectively). The pitch angle is also a function of inflow wind speed and depends on the control scheme that is used. We determined the pitch angles using CCBlade at several wind speeds (above the rated wind speed) and a zero finder to calculate the pitch angle necessary to provide the rated power at each wind speed, assuming a generator efficiency of 93%. The pitch angle as a function of wind speed is shown in Fig. 3. Note that we manually applied a slight smoothing to the pitch
function near the rated wind speed to facilitate optimization.

After calculating the loads distributions along the blade, we calculated the bending moments by integrating the loads along the length of the blade, shown in Eq. 2. In this paper, we have assumed that the blade root is the fatigue critical location.

$$M = \int_{0}^{R_{\text{tip}}} q(r)r \ dr \tag{2}$$

The bending moment, $M$, is caused by the loading along the blade, $r$ is the distance along the blade, and $q$ is the loading along
the blade. In this step, it is important to keep the coordinate systems straight. For example, CCBlade returns the flapwise and lead-lag forces on the blade; therefore, $q$ was adjusted by the blade pitch.

**Table 1.** Tuning constants for the root bending moment surrogate functions. The azimuth angle and the blade pitch are represented by $\psi$ (in degrees) and $\theta$ (in radians), respectively.

|  | a (N m) | b (N m) | c (N m)/rad | d rad | e unitless | g (N m) |
|---|---|---|---|---|---|---|
| flatwise, $\psi = 90°$ | 9110 | 942 | 48000 | - | - | - |
| flatwise, $\psi = 270°$ | 9110 | 762 | 40000 | - | - | - |
| edgewise, $\psi = 90°$, $\theta < 0.05$ | 1540 | 406 | 43000 | 0.05 | 2 | 100.0 |
| edgewise, $\psi = 90°$, $\theta \geq 0.05$ | 1540 | 406 | 43000 | 0.05 | 1.85 | 100.0 |
| edgewise, $\psi = 270°$, $\theta < 0.05$ | 1540 | 390 | 29500 | 0.05 | 2 | 75.0 |
| edgewise, $\psi = 270°$, $\theta \geq 0.05$ | 1540 | 390 | 29500 | 0.05 | 1.8 | 75.0 |

Now, with the root bending moments for different wind speeds and pitch angles calculated, we fit a piecewise function to the data to create a set of simple relations.

$$M = \begin{cases} a\left(\frac{U_{\text{rotor}}}{U_{\Omega\text{max}}}\right)^2 - f(\theta) & U_{\text{rotor}} \leq U_{\Omega\text{max}} \\ a + b(U_{\text{rotor}} - U_{\Omega\text{max}}) - f(\theta) & U_{\text{rotor}} > U_{\Omega\text{max}} \end{cases} \tag{3}$$

In this equation, $U_{\text{rotor}}$ is the inflow wind speed to the rotor, $U_{\Omega\text{max}}$ is the wind speed at which the maximum rotation speed occurs, which is 10.62 m/s for the reference turbine we used, the $a$ and $b$ are constants, and $f(\theta)$ is a function representing how the moments decrease with pitch angle:

$$f(\theta) = \begin{cases} c\,\theta & \text{flatwise moments} \\ c(\theta - d)^e + g & \text{edgewise moments} \end{cases} \tag{4}$$

where $\theta$ is the blade pitch angle in radians, and $c$, $d$, $e$, and $g$ are constants. Because the turbine used in this paper had a non-zero tilt, the tuning constants in Eqs. 3 and 4 vary slightly depending on the blade azimuth angle. Additionally, the tuning constants vary depending on the moment of interest. These constants are given in Table 1. Figure 4 shows our surrogate fit to higher fidelity data. As seen in the figure, the surrogate fits very well through about 18–20 meters per second, where there is a slight differential near the cut-out speed of 25 meters per second. For the purposes of this paper, our wind resources were much lower than the cut-out speed, so this was not an issue. If one were to include higher wind speeds or hourly wind speed data, an improved surrogate may be beneficial.

We have presented a simple piecewise surrogate that will be shown to predict the fatigue damage sufficiently well. However, it is important to remember that there are many potential models that could be used to predict the blade root bending moments. For our surrogate model, it is important that we mention a couple of items that may not be immediately obvious. First, in Eq. 3, there is a pitch modification to the loads for the wind speeds acting on the blades above and below $U_{\Omega\text{max}}$. The reason that the pitch modification is included for wind speeds below $U_{\Omega\text{max}}$ is because the blade pitch angle is determined by the wind speed across the entire rotor, while the load surrogate is a function of the wind speed for a single blade. In cases of partial waking,

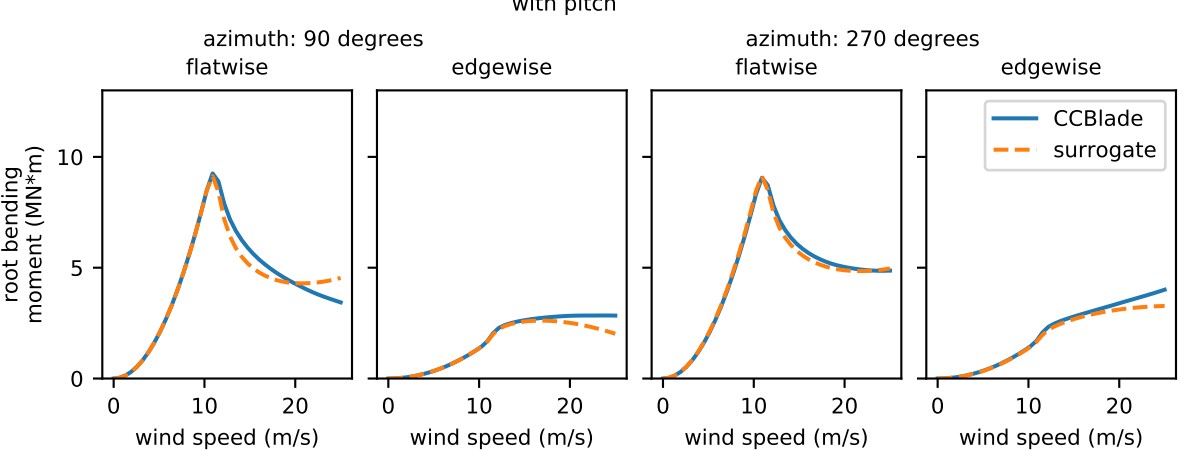

**Figure 4.** The surrogate fit for the root bending moments. From left to right, the $R^2$ value of each fit is 0.966, 0.939, 0.995, 0.967, respectively.

it is possible for the wind speed for a single blade to be relatively low, while the speed acting across the entire rotor is much higher, which would cause the blades to pitch. Second, when calculating the moments for our surrogate, we have assumed that the rotation speed of the rotor is a function of the blade wind speed, instead of the rotor wind speed. This greatly simplified the surrogate model and is an acceptable assumption because the loads are much less sensitive to the rotation speed than they are to the pitch angle.

### 2.3 Calculate Steady-State Turbine Inflow Wind Speed

After defining the turbulence samples and loads surrogates, the next step is to calculate the steady-state turbine inflow wind speed. This is done with an analytic wake model used to predict wind speeds in a wind farm. For this paper, we found good results with a modified Gaussian wake model presented by Bastankhah and Porté-Agel (Bastankhah and Porté-Agel, 2016). Overall, this model performs very well at capturing the velocity profile in the wake of a turbine, matching high fidelity data very well. For our purposes, the most important physical effects that this model does not capture is inflow flow heterogeneity, which can affect power production and loads. The original formulation of the model does not define the wake deficit in the near wake region. This near-wake region, called the potential core, results in regions behind wind turbines where the wind speed is undefined. These undefined regions in the space make optimization difficult because the objective function is undefined in some places. To mitigate this issue, Thomas and Ning added a linear interpolation of the wake loss from the turbine up to where it is defined by the wake model, which is the version used in this paper (Thomas and Ning, 2018). The most important equation for this Gaussian wake model is shown in this equation:

$$\frac{\Delta U}{U_\infty} = \left(1 - \sqrt{1 - \frac{C_T \cos\gamma}{8\sigma_y \sigma_z / d^2}}\right) \exp\left(-0.5\left(\frac{y - \delta}{\sigma_y}\right)^2\right) \exp\left(-0.5\left(\frac{z - z_h}{\sigma_z}\right)^2\right) \tag{5}$$

where $\Delta U/U_\infty$ is the velocity deficit at a point in the wake; $C_T$ is the thrust coefficient; $\gamma$ is the yaw angle, which is assumed to be zero throughout this paper; $y - \delta$ and $z - z_h$ are the distances from the wake center and the point of interest in the cross-stream horizontal and vertical directions, respectively (where $\delta$ is the wake center which can be deflected with wake steering, and $z_h$ is the hub height of the turbine creating the wake); and $\sigma_y$ and $\sigma_z$ are the standard deviations of the wake deficit, again in the cross-stream horizontal and vertical directions, respectively, and are shown in the following equations:

$$\sigma_y = k_y(x - x_0) + \frac{D \cos \gamma}{\sqrt{8}} \tag{6}$$

$$\sigma_z = k_z(x - x_0) + \frac{D}{\sqrt{8}} \tag{7}$$

In these equations, $D$ is the diameter of the wind turbine creating the wake, $x - x_0$ is the distance downstream from the end of the potential core to the point of interest, and $k_y$ and $k_z$ are unitless, and are functions of the freestream turbulence intensity:

$$k_y, k_z = c_1 \text{TI} + c_2 \tag{8}$$

In Eq. 8, $c_1$ and $c_2$ are tuning parameters, while TI is the turbulence intensity on the upstream wind turbine. The length of the potential core, $x_0$, is defined in Eq. 9.

$$x_0 = \frac{D \cos \gamma (1 + \sqrt{1 - C_T})}{\sqrt{2}[\alpha \text{TI} + \beta (1 - \sqrt{1 - C_T})]} \tag{9}$$

With the correct tuning parameters, $c_1$, $c_2$, $\alpha$, and $\beta$, we were able to approximate velocity data from SOWFA with the analytic wake model. Note that because the yaw angle, $\gamma$, is assumed to be zero throughout this paper, there is no wake steering so the wake center in the y-coordinate, $\delta$, is the same as the y coordinate of the turbine from which the wake originates, and $\cos(\gamma) = 1$, meaning that $\sigma_y = \sigma_z$.

To calculate the loads on a wind turbine in this study, we needed to be able to accurately predict the local turbulence intensity throughout the wind farm. Additionally, for this wake model we need the turbulence intensity for the inflow into a turbine. We used two different models to fit these two requirements. The model to calculate local turbulence intensity will be discussed in Section 2.4, while the algorithm to calculate the inflow turbulence intensity to a turbine is provided in Thomas et al. (Thomas et al., 2019).

Figure 5 shows the velocity profiles predicted by the wake model compared to the time average velocity data from our SOWFA runs for 4, 7, and 10 diameters downstream of a wind turbine and for two different freestream turbulence intensities. The y-axes give the wind speed, while the x-axes indicate the cross-stream offset (in rotor diameters) from the center of the turbine generating the wake. Each column shows the wake profiles for a different distance downstream (in rotor diameters), while the left and right halves of this figure gives wake profiles for low and high ambient turbulence intensities, respectively. Note that the model has been tuned for each wind speed. For the low-turbulence-intensity cases, there is good agreement between the model and the SOWFA data for every wind speed, with the largest errors occurring for the freestream wind speed of 13 m/s at 4 and 7 rotor diameters downstream of the waking turbine. For the higher-turbulence-intensity cases, the model

**Table 2.** Tuning constants for the velocity data shown in Fig. 5

| wind speed (m/s) | freestream TI (%) | $c_1$ | $c_2$ | $\alpha$ | $\beta$ | $R^2$ |
|---|---|---|---|---|---|---|
| 10 | 4.6 | 0.0908 | 0.00831 | 3.78 | 0.735 | 0.863 |
| 11 | 4.6 | 0.0629 | 0.00693 | 2.64 | 0.457 | 0.954 |
| 12 | 4.6 | 0.0114 | 0.00805 | 3.73 | 0.132 | 0.912 |
| 13 | 4.6 | 0 | 0 | 3.66 | 0 | 0.908 |
| 10 | 8 | 0.121 | 0.00127 | 10.0 | 10.0 | 0.782 |
| 11 | 8 | 0.0000765 | 0.0390 | 3.79 | 0.154 | 0.624 |
| 12 | 8 | 0.0382 | 0.0164 | 3.85 | 0.585 | 0.874 |
| 13 | 8 | 0.823 | 0.158 | 3.49 | 0.356 | 0.249 |

does a decent job of predicting the wind speed deficits, meaning the difference between the freestream speeds and the minimum speeds. The high-turbulence SOWFA data has areas of local velocity differences, which are not captured in our analytic wake model. The model does not capture the asymmetry in the wake and underpredicts the velocity magnitudes by about 1 m/s in many locations. The model performs the worst for the 11 m/s and 13 m/s high-turbulence cases, although it does a decent job at capturing the deficit. The data for these cases are hard to match because the wake is highly asymmetric. The tuning parameters used for the SOWFA data shown in the figure are provided in Table 2. In addition to the tuning constants, this table also shows the $R^2$ value of the model fit to the SOWFA velocity data. As can be seen, the fits for the lower turbulence scenarios are much better than for the higher turbulence ones.

In the case of combined wakes, the total wind speed was calculated with a linear combination method.

$$U = U_\infty - \sum_{i=1}^{\text{nTurbs}} U_i \left( \frac{\Delta U}{U_\infty} \right)_i \tag{10}$$

In this equation, $U$ is the local wind speed at a given point, $U_\infty$ is the freestream wind speed, nTurbs is the number of wind turbines upstream of the point of interest, $U_i$ is the inflow speed of an upstream rotor, and $\left( \frac{\Delta U}{U_\infty} \right)_i$ is the velocity deficit from an upstream turbine. This wake combination method has been shown to compare well with experimental data when combined with the Gaussian wake model we used (Niayifar and Porté-Agel, 2016). Additionally, this combination method is equally computationally efficient wake combination methods, such as taking the two-norm of the wake deficits.

The wake model above has all been defined to calculate the wind speed at a given point. To determine the wind speed into a wind rotor, we took the average of the wind speed calculated at four points across the rotor, shown in Fig. 6. Also shown in Fig. 6 is the average wind speed profile to a downstream turbine as it is moved across the wake of an upstream turbine. As seen, the average wind speed profile with four sample points is graphically similar to 100 sample points, while 1 sample point is not sufficient. In addition to the wind speed inflow, we found that sampling at these four points gives almost identical damage values as sampling with many more points across the rotor. Figure 7 shows the final damage values for one turbine as a function of cross-stream offset at different distances downstream of a waking turbine. This offset is given in rotor diameters,

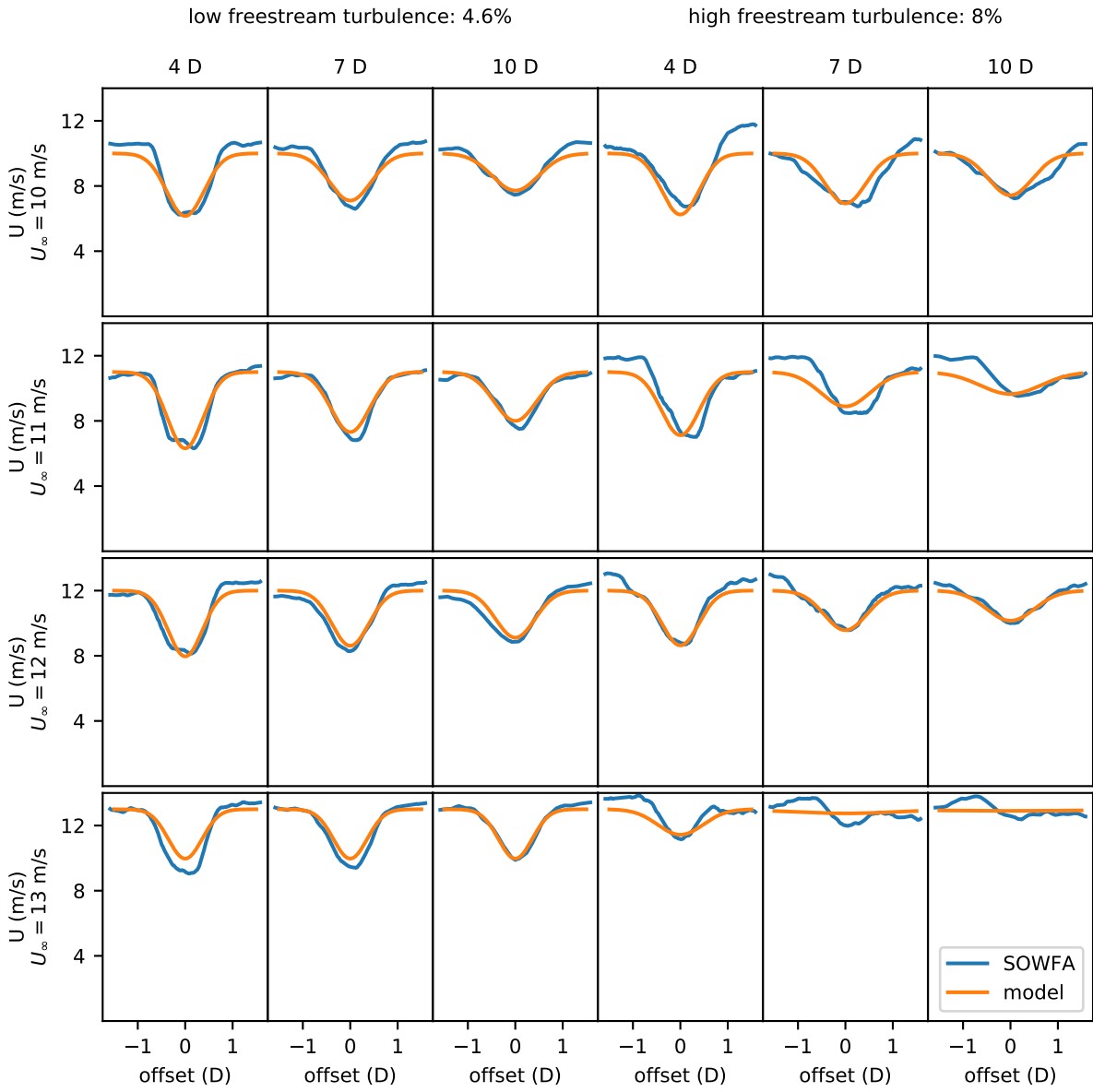

**Figure 5.** The wind speeds predicted by our large eddy simulation data compared to the analytic wake model for freestream turbulence intensities of 4.6% (3 left columns) and 8% (3 right columns). From left to right, each column represents the wind speeds at 4, 7, and 10 D downstream of the turbine generating the wake. From top to bottom, each row represents the wind speeds with a freestream wind speed of 10, 11, 12, and 13 m/s.

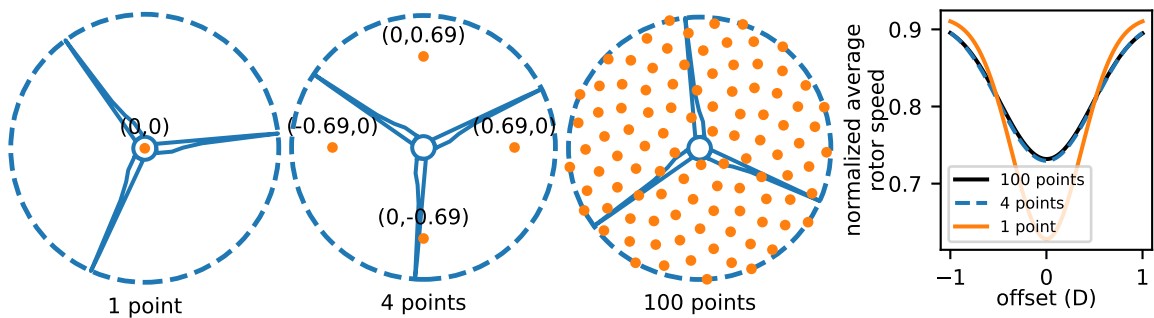

**Figure 6.** The sample points used to calculate the average wind speed into a turbine. From left to right, the location of the 1, 4, and 100 sample points are shown on the swept area of the rotor. On the far right: the average rotor speed of a turbine moving across the wake of an upsteam turbine. The downstream turbine moves from one side of the wake (-1 D offset) to the other side of the wake (1 D offset). Each of the 3 lines shows the average rotor wind speed calculated with 1, 4, and 100 sample points.

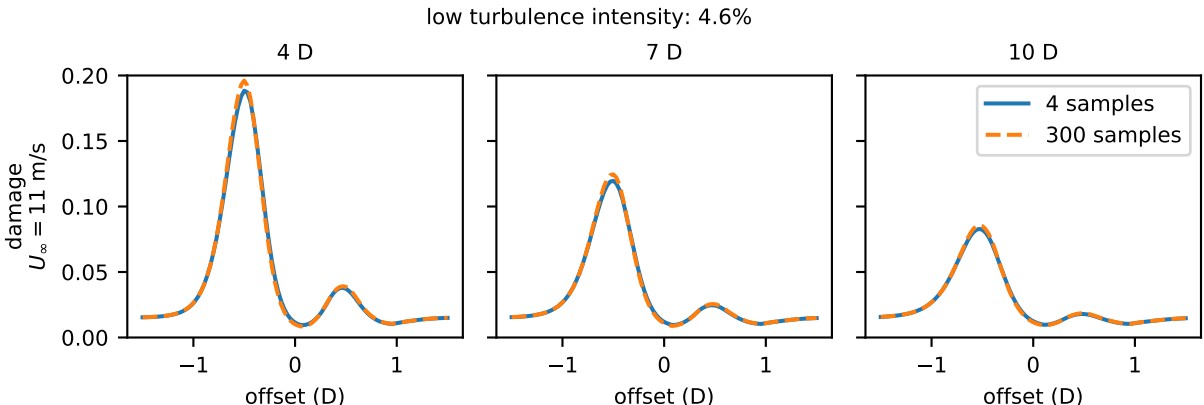

**Figure 7.** Final damage plots calculated with different numbers of rotor samples to determine the average wind speed across the rotor. The blue line represents damage calculated with 4 sample points, the orange line represents the damage calculated with 300 sample points. Notice they are almost identical.

and indicates the position of the downstream turbine hub relative to the upstream turbine hub. The figure indicates that the damage values with our four sample points is graphically similar to damage calculated with many more points to determine the averaged inflow wind speed to the rotor. The damage values shown in this figure are calculated with the model that is fully presented in the rest of this section. Even though the full details of this model are presented in the subsections below, we determined that it is appropriate to present this information here to demonstrate the minimal effect that the number of wind speed samples across the swept rotor area has on the final result.

This section has discussed in detail the analytic wake model we used in this paper. Remember that the fatigue model that we will present does not require this specific wake model. However, it does depend on the ability to provide accurate wind speeds

for various locations throughout the wind farm. We found success with the wake model presented in this section; however, other wake models or methods to calculate the wind speed may also be used. That said, the damage model is particularly sensitive to the mean velocity profiles predicted by the wake model, which is why we tuned each wind speed individually to best match the velocities while demonstrating our damage model.

## 2.4 Calculate Turbulence Intensities Across the Blade

The next step in the model is to calculate the turbulence intensity affecting the blade at each azimuth angle, which is defined as the standard deviation of the streamwise wind speed divided by the undisturbed mean. This requires the ability to accurately calculate the turbulence intensity at any given location. To accomplish this, we used a modified version of the model presented by Ishihara and Qian (Ishihara and Qian, 2018).

$$\Delta\text{TI}(x,y,z) = \frac{1}{d + e \cdot x/D + f \cdot (1 + x/D)^{-2}} \cdot \left\{ k_1 \exp - \frac{(r - D/2)^2}{2\sigma_t^2} + k_2 \exp - \frac{(r + D/2)^2}{2\sigma_t^2} \right\} - \delta(z) \tag{11}$$

The added turbulence intensity $\Delta\text{TI}$ is caused by an upstream wind turbine, $x$ and $y$ are the downstream and cross-stream distances from the point of interest to the upstream turbine, $z$ is the height of the point of interest, $\delta(z)$ accounts for weakened turbulence closer to the ground as defined in , and $D$ is the rotor diameter of the upstream turbine. The equation for $\delta(z)$ and the rest of the values in Eq. 11 are given in the following equations:

$$\delta(z) = \begin{cases} 0 & z >= z_h \\ \text{TI}_a \sin^2(\pi(z_h - z)/z_h) & z < z_h \end{cases} \tag{12}$$

$$d = 2.3 C_T^{-1.2} \tag{13}$$

$$e = \text{TI}_a^{0.1} \tag{14}$$

$$f = 0.7 C_T^{-3.2} \text{TI}_a^{-0.45} \tag{15}$$

where $C_T$ is the thrust coefficient of the upstream wind turbine, and $\text{TI}_a$ is the ambient turbulence intensity. The radial distance to the point of interest, $r$, is given by Eq. 16,

$$r = \sqrt{y^2 + (z - H)^2} \tag{16}$$

where $H$ is the hub height of the upstream turbine. The values for $k_1$ and $k_2$ are given in Eqs. 17 and 18.

$$k_1 = \begin{cases} \cos^2\left(\pi/2 \cdot (r/D - 0.5)\right) & r/D \leq 0.5 \\ 1 & r/D > 0.5 \end{cases} \tag{17}$$

$$k_2 = \begin{cases} \cos^2\left(\pi/2 \cdot (r/D + 0.5)\right) & r/D \leq 0.5 \\ 0 & r/D > 0.5 \end{cases} \tag{18}$$

Finally, $\sigma_t$ is a representative wake width for the local turbulence-intensity model, shown in the following equations.

$$\sigma_t/D = k^* x/D + \epsilon \tag{19}$$

$$k^* = 0.11 C_T^{1.07} \mathrm{TI}_a^{0.20} \tag{20}$$

$$\epsilon = 0.23 C_T^{-0.25} \mathrm{TI}_a^{0.17} \tag{21}$$

As formulated, this turbulence-intensity model has a wide wake and predicts a large turbulence intensity increase behind a wake. To better match our SOWFA data, we made some slight adjustments to the model by introducing two tuning parameters, $C_1$ and $C_2$, which change Eqs. 11 and 19. These constants divide the wake width and turbulence intensity increases predicted by the model.

$$\Delta\mathrm{TI}(x,y,z) = \frac{1}{C_1}\left(\frac{1}{d + e \cdot x/D + f \cdot (1+x/D)^{-2}} \cdot \left\{k_1 \exp{-\frac{(r-D/2)^2}{2\sigma_t^2}} + k_2 \exp{-\frac{(r+D/2)^2}{2\sigma_t^2}}\right\} - \delta(z)\right) \tag{22}$$

$$\sigma_t/D = \frac{1}{C_2}(k^* x/D + \epsilon) \tag{23}$$

Figure 8 shows the predicted turbulence intensity from the model compared to our SOWFA data. This figure shows a sweep of the turbulence intensity across the turbine wake at hub height. The three columns on the left show the model comparison for a low freestream turbulence of 4.6%. There is good comparison between the analytic model, representing the trends and actual values of the local turbulence intensities. The three columns on the right show the model comparison for a high freestream turbulence of 8%. With high freestream turbulence, the analytic model does not match the SOWFA data as well. The local turbulence intensity does not follow the same structure as the model predicts, but at least reasonably captures the magnitudes. As will be seen later, our fatigue damage model does not match the high-fidelity data as well for high-turbulence-intensity cases. In part, this is likely due to the mismatch between the turbulence model and the turbulence SOWFA data. The turbulence-intensity model represented in these figures has been tuned for each freestream wind speed and turbulence intensity, for which the tuning constants are listed in Table 3. As with the table showing the tuning constants for the velocity data, this table also shows the $R^2$ value of the model fit to the SOWFA turbulence data. Notice that the fit is no as good as for the velocity model shown previously. The SOWFA turbulence data is very noisy, and in some of the cases does not even have a clear coherent

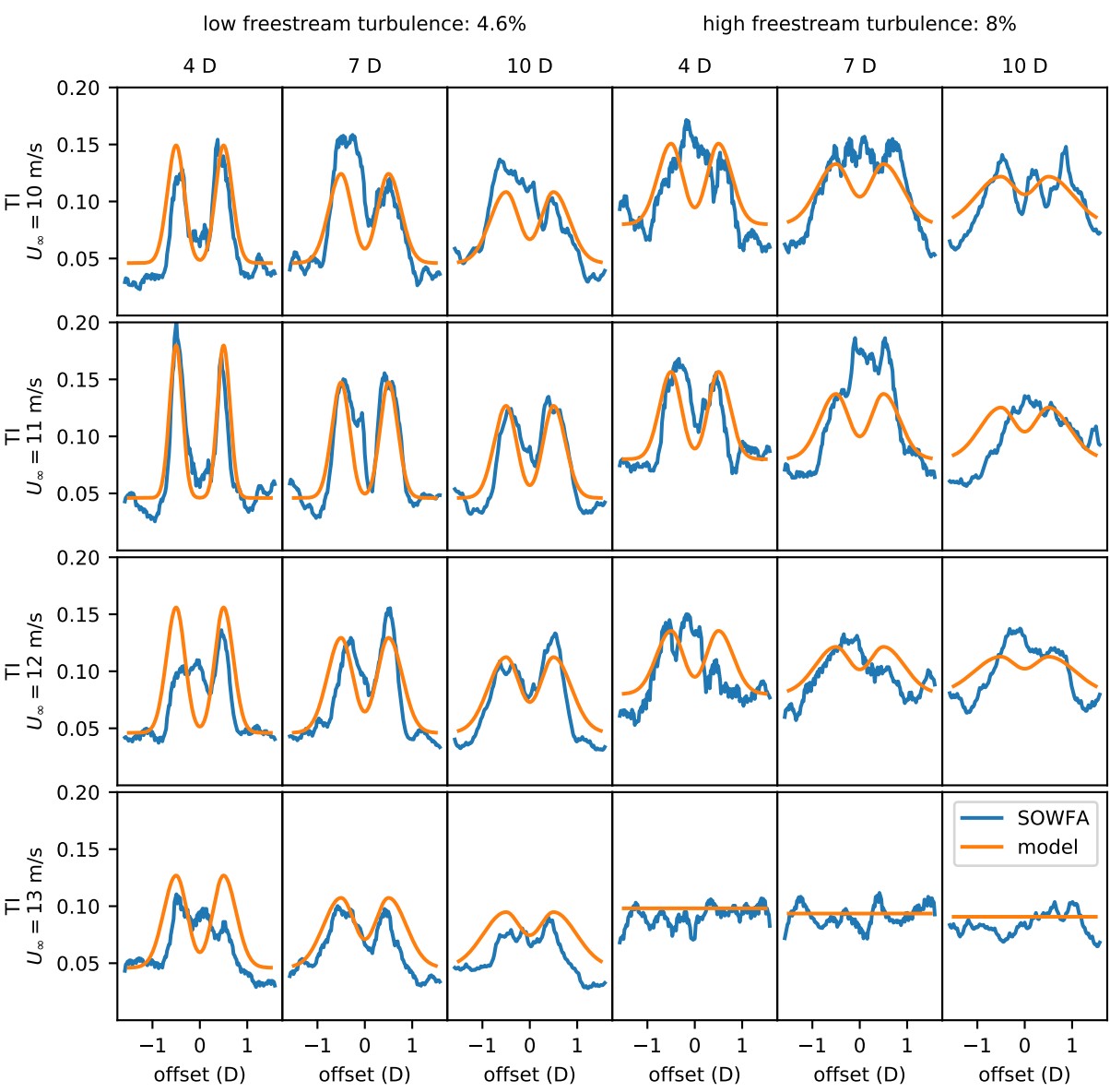

**Figure 8.** The turbulence intensity predicted by our large eddy simulation data compared to the analytic turbulence-intensity model for freestream turbulence intensities of 4.6% (3 left columns) and 8% (3 right columns). From left to right, each column represents the turbulence intensities at 4, 7, and 10 D downstream of the turbine generating the wake. From top to bottom, each row represents the turbulence intensities with a freestream wind speed of 10, 11, 12, and 13 m/s.

**Table 3.** Tuning constants for the turbulence data shown in Fig. 8

| wind speed (m/s) | freestream TI (%) | $C_1$ | $C_2$ | $R^2$ |
|:---:|:---:|:---:|:---:|:---:|
| 10 | 4.6 | 1.51 | 1.75 | 0.542 |
| 11 | 4.6 | 1.17 | 2.35 | 0.706 |
| 12 | 4.6 | 1.42 | 1.58 | 0.527 |
| 13 | 4.6 | 1.93 | 1.22 | 0.468 |
| 10 | 8 | 2.17 | 1.27 | 0.417 |
| 11 | 8 | 2.01 | 1.48 | 0.323 |
| 12 | 8 | 2.77 | 1.16 | 0.291 |
| 13 | 8 | 8.49 | 0 | 0.0590 |

structure. This makes it difficult for the analytic model to fit, although it does a good job at predicting the overall trends and relative magnitudes.

With the turbulence-intensity model defined, the average turbulence intensity over the entire blade can be calculated. This is done by integrating the turbulence intensity over the length of the blade.

$$\text{TI}_{\text{blade}} = \frac{1}{R_{\text{tip}}} \int_0^{R_{\text{tip}}} \text{TI}(r) \, dr \tag{24}$$

In this equation, $\text{TI}_{\text{blade}}$ is the turbulence intensity that acts over the length of the blade, $R_{\text{tip}}$ is the radius of the blade at the tip, and TI is the local turbulence intensity evaluated along the length of the blade, $r$, where $\text{TI}(r) = \text{TI}_a + \Delta\text{TI}$ and $\Delta\text{TI}$ is given in Eq. 22. This turbulence intensity for the blade is evaluated for each azimuth angle that is being considered.

In addition to the turbulence intensity for the blade, we also calculate a turbulence intensity for the entire rotor, $\text{TI}_{\text{rotor}}$. This is defined as the average of the blade turbulence intensities calculated at each azimuth angle.

## 2.5 Calculate Blade Wind Speeds

In addition to calculating the turbulence intensity across the blade at each azimuth angle that is considered, our model also requires the calculation of the wind speed acting along the blade. This was accomplished by using the same wake model described in Sec. 2.3, Eq. 10 and integrating the wind speed along the length of the blade. This acting blade wind speed is used later to calculate the moments on the turbine blade while considering turbulence. The blade wind speed is defined as the integral of the wind speed along the blade, as shown in Eq. 25.

$$U_{\text{blade}} = \frac{1}{R_{\text{tip}}} \int_0^{R_{\text{tip}}} U(r) \, dr \tag{25}$$

## 2.6 Turbulence and Azimuth Loop

The three sections following this one discuss steps 7, 8, and 9. These steps occur in a loop, which creates a load history that accounts for the different azimuth angles of the blade and the different loading that occurs at each rotation caused by turbulence. The steps in this loop are to: 1.7) calculate the turbine inflow wind speed accounting for turbulence, 1.8) using this inflow speed, determine the turbine rotational speed and blade pitch, and 1.9) determine average turbulent wind speed across a blade, and use this speed and the blade pitch in the loads surrogate to determine the blade loads at the time step. These steps are then repeated for as many azimuth angles and rotations that will be simulated. After each time through the loop, the loads calculated in step 1.9 are added to a loads history. The end result is a history of the flatwise and edgewise blade loads, which is used in future steps to make fatigue calculations.

For all of the cases that we tested, we used two azimuth angles of 90 degrees and 270 degrees to predict the fatigue damage. These azimuth angles correspond to when the turbine blade is parallel to the ground on opposite sides of the rotor. At these angles, the gravitational loading is at the extreme values. Additionally, the load variations caused by partial waking are the largest between these two azimuth angles. In some cases with high wind shear, it may be appropriate to also include azimuth angles of 0 and 180 degrees, between which the differences in wind speed due to wind shear are the largest. However, at these angles, the moments due to gravity are zero, indicating that the flatwise loads would need to be very large to introduce larger load fluctuations than would occur at azimuth angles of 90 and 270 degrees. In reality, there are a multitude of small fluctuations that occur throughout the entire rotor rotation.

Figure 9 shows the final damage results calculated with 2 azimuth angles and 100 azimuth angles, as well as damage from the high-fidelity SOWFA simulation. To calculate these damages, we directly evaluated the blade loads using CCBlade, instead of a surrogate model as discussed in Sec. 2.2. Notice the differences between the damage predictions for 2 and 100 azimuth angles. The damage with 100 azimuth angles is slightly higher at the peak at 4 and 7 rotor diameters downstream of the waking turbine. However, they are relatively close to each other, both capture the same trends of minimum and maximum damage, and are follow the high-fidelity damage prediction remarkably well. The computational expense of our damage model scales linearly with the number of azimuth angles that are modeled. Because two azimuth angles compared so well with more azimuth angles and with the the SOWFA data, we determined that considering just two azimuth angles was sufficient as it captured the largest load differences which contribute the most to fatigue damage and minimized the computational expense. As with Fig. 7, the full details of the fatigue damage model are presented throughout this entire section. Even though part of this model is described below this figure, we think it is appropriate to present this figure here to show the effect that the number of azimuth angle evaluations has of the final damage results.

## 2.7 Calculate Turbulent Turbine Inflow Wind Speed

Once in the loop, the first step is to calculate the average turbine inflow wind speed with turbulence.

$$\text{TI} = \frac{\sigma_u}{\bar{u}} \tag{26}$$

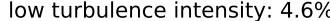

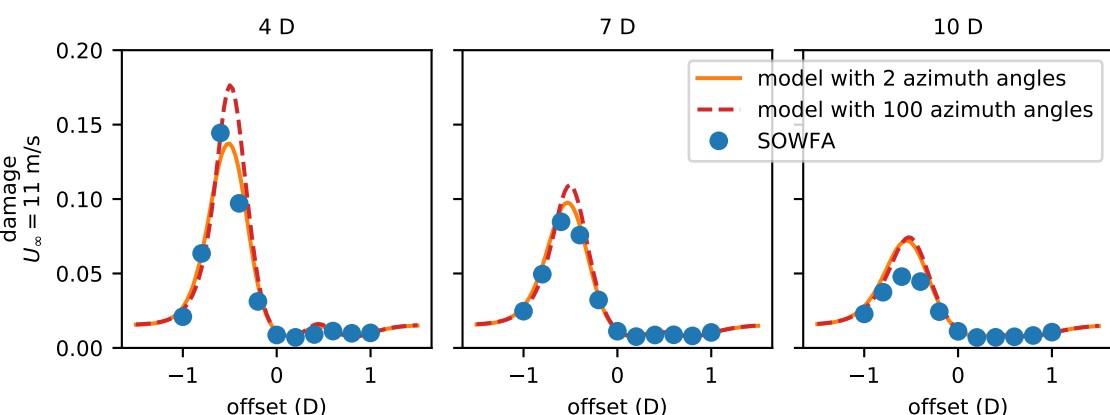

**Figure 9.** Final damage plots calculated with different numbers of azimuth angle evaluations. The orange line represents damage calculated with 2 azimuth angle evaluations at 90 and 270 degrees, the dashed red line represents the damage calculated with 100 azimuth angle evaluations, and the blue points show the damage calculated with the high-fidelity SOWFA data.

In this equation, $\sigma_u$ is the standard deviation of the streamwise wind speed, and $\bar{u}$ is the mean wind speed at the given point. Using this definition of turbulence intensity, we defined the instantaneous turbine inflow velocity:

$$u_{\text{turbulent, rotor}} = U_{\text{rotor}}(1 + S_i\,\text{TI}_{\text{rotor}}) \tag{27}$$

In this equation, $u_{\text{turbulent}}$ is the instantaneous turbine inflow that accounts for turbulence, $U_{\text{steady}}$ is the steady state turbine inflow velocity calculated in step 3, and $S_i$ is the turbulence sample corresponding to the azimuth angle and rotation being calculated, which was defined in step 1. The turbulence intensity $\text{TI}_{\text{rotor}}$ is the acting rotor TI value calculated in step 4. These values were discussed previously because they only needed to be calculated once, and not within the turbulence and azimuth angle loop.

## 2.8 Calculate Rotational Speed and Pitch Angle

Using the control scheme of the turbine being modeled, the rotational speed of the turbine is calculated and stored based on the turbulent turbine inflow wind speed calculated previously. The instantaneous pitch angle is also calculated, which will be used when finding the instantaneous bending moments.

## 2.9 Calculate Turbulent Bending Moments

This next critical step is to calculate the bending moment history with the instantaneous wind speeds that take turbulence into account. This could be done directly, by calling CCBlade within the turbulence and azimuth angle loop to get the blade loads, then converting them into bending moments. However, this is unnecessarily expensive. In a single analysis, taking a few extra

seconds to use a complete model is worth the small extra computation time. However, in an optimization framework, the model must be called thousands or tens of thousands of times to find a solution. In this case, slightly sacrificing accuracy by using a surrogate model is well worth the savings in computational expense. Sufficiently accurate bending moments can be calculated by using the surrogate we created in step 2. This is done by taking the blade wind speed calculated in step 5 and scaling it with the turbulence, shown in Eq. 28.

$$u_{\text{turbulent, blade}} = U_{\text{blade}}(1 + S_i \, \text{TI}_{\text{blade}}) \tag{28}$$

This turbulent blade wind speed and the pitch angle from step 8 are then used to evaluate the bending moment surrogate models. These moments evaluated with the surrogate model only consider the aerodynamic loads and do not take gravity into account, which we must also include.

$$
\begin{aligned}
M_{\text{gravity}} &= m \, g \, r_{\text{cm}} \sin\psi \cos\phi \cos\Theta \\
M_{\text{flatwise}} &\mathrel{+}= M_{\text{gravity}} \sin\theta \\
M_{\text{edgewise}} &\mathrel{+}= M_{\text{gravity}} \cos\theta
\end{aligned}
\tag{29}
$$

In this equation, $m$, $g$, and $r_{\text{cm}}$ represent the blade mass, gravitational constant, and radial location of the center of gravity of the blade. For the reference turbine we used, these values are given by 17,537 kg, 9.81 m/s$^2$, and 20.65 m, respectively (Resor, 2013). The other values, $\psi$, $\phi$, $\Theta$, and $\theta$, represent the azimuth angle, precone, tilt, and blade pitch angle, respectively. For the reference turbine, the precone was 2.5 degrees and the tilt was 5 degrees.

## 2.10 Radial Damage Location Loop

After completing the turbulence and azimuth angle loop, the moment history is complete. However, the fatigue damage is dependent on the stress history, which is calculated from the moment history in the next step. The stress depends on how the flatwise and edgewise moments interact at each load cycle and is also different depending on the location around the circumference of the blade root where the stress is calculated. Without knowing the stress history, it is impossible to know beforehand where the location of maximum damage will be. Thus, to make sure we calculate the highest fatigue value experienced by a turbine for a given loading condition, we calculated the stress history and the associated damage at several locations around the circumference of the blade root. Because exact opposite sides of the blade root experience the same stress cycle, except with the sign flipped, we only considered locations around one half of the blade root. The results shown in this paper were done with 50 stress location samples evenly spaced around half of the circumference.

## 2.11 Convert Bending Moments to Stresses

Before calculating the damage, the moment history must be converted to a stress history at the location of interest. This step, along with the next, is done in a loop for each location around the circumference of the blade root. Finding the moments is a

simple conversion (Budynas and Nisbett, 2020).

$$\sigma_y = -\frac{M_z x}{I_z} + \frac{M_x z}{I_x} \tag{30}$$

In this equation, $\sigma_y$ is the stress at the blade root; $M_x$ and $M_z$ are the moments about the $x$ and $z$ axes; respectively, $x$ and $z$ are the distances from the center of the blade root to the location of interest in the $x$ and $z$ directions; and $I_x$ and $I_z$ are the second moments of inertia about the $x$ and $z$ axes, respectively. When calculating the stresses, one must be careful to use a consistent coordinate system, such that when the blade is pitched, the stress is still calculated in the same location. We assume the blade root is a hollow cylinder, for which the moment of inertia is given in Eq 31.

$$I_x = I_z = \frac{\pi}{4}(R_{\text{outer}}^4 - R_{\text{inner}}^4) \tag{31}$$

In this equation, $R_{\text{outer}}$ represents the outer radius of the blade root, and $R_{\text{inner}}$ represents the inner radius. For the NREL 5-MW reference turbine, these values are 1.693 meters and 1.643 meters, respectively (Resor, 2013). Note that for these equations we are using axes where the $x$ axis is in the freestream wind direction, the $y$ axis is along the blade, and the $z$ axis is the direction of blade rotation. After testing, we found that the contribution from shear was negligible compared to the bending moments because of the large moment arms. As seen in Eq. 30, we ignored the contributions from the shear forces in the stress calculations.

## 2.12 Calculate Damage

From the stress history, the damage accumulated by a wind turbine throughout its lifetime is calculated for the given load conditions. First, rainflow counting was used to determine all of the stress cycle ranges and peaks. Rainflow counting is a commonly used method to extract all of the loading cycles that occur in a noisy set of data (Matsuishi and Endo, 1968). A Goodman correction was then applied to account for the mean loading effects and extract an equivalent fully reversed load:

$$\sigma_{er} = \frac{\sigma_a}{1 - \sigma_m/\sigma_U} \tag{32}$$

where $\sigma_{er}$ is the effective fully reversed stress amplitude, $\sigma_a$ is the stress amplitude for a given stress cycle, $\sigma_m$ is the mean stress of the stress cycle, and $\sigma_U$ is the material ultimate stress, which was assumed to be 350 MPa at the blade root. The material described in the reference turbine blade has an ultimate stress of 700 MPa, which is given in tension. We assumed the ultimate stress in compression would be lower and applied a small knockdown factor to arrive at 350 MPa (Resor, 2013). The cycles to failure for each effective fully reversed load were then calculated as shown in mLife, a wind turbine fatigue calculation code (Hayman, 2012):

$$N_{\text{fail}} = \left(\frac{\sigma_U}{\sigma_{er}\,SF}\right)^m \tag{33}$$

where $N_{\text{fail}}$ is the number of cycles to failure, $SF$ is a safety factor, and $m$ is the material-dependent Wöhler exponent. For composite turbine blades, it is typically assumed that $m = 10$, which is the value used in this study (Ingersoll and Ning, 2018).

Miner's rule was then used to calculate the damage accumulated by a turbine over a 25-year lifespan, shown in Eq. 34:

$$d = \frac{N_{\text{cycles},i}}{N_{\text{fail},i}} \tag{34}$$

where $d$ is the damage accumulated by the blade at the specified location around the blade circumference and $N_{\text{cycles},i}$ is the number of cycles that the blade experiences at the given loading condition. The number of cycles a blade would experience at a given condition over its lifetime is defined in Eq. 35:

$$N_{\text{cycles},i} = \frac{86400 \cdot 365.35 \cdot P_i \, N_{\text{years}} \, N_{\text{count}}}{t_{\text{simulated}}} \tag{35}$$

where 86,400 is the number of seconds in a day; 365.25 is the number of days in a year; $P_i$ is the probability of the loading condition occurring; $N_{\text{years}}$ is the desired lifetime of the wind turbine, which was assumed to be 25 years for this study; $N_{\text{count}}$ is the number of times the given loading condition happened during the simulation (this was extracted with the rainflow counting); and $t_{\text{simulated}}$ is the total time of the simulation. Equation 36 defines $t_{\text{simulated}}$.

$$t_{\text{simulated}} = N_{\text{cycles}} \frac{2\pi}{\Omega} \tag{36}$$

In this equation, $\Omega$ is the average of the rotor rotation speed calculated in step 8, and $N_{\text{cycles}}$ is the number of rotor rotations included in the simulation, which was 50 for the results shown in this paper. Although conventional time domain-based fatigue estimates are generally based on a longer time period of simulation, for the purposes of this paper in which we demonstrate the use of this proposed model, this shorter time was used for decreased computational expense.

## 2.13 Return Maximum Damage

Finally, after calculating the fatigue damage at each of the locations around the blade root circumference, return the maximum damage value. We tested the model for a variety of loading conditions and found that, for the situations that we tested, the locations of maximum damage were all within ten degrees of each other. Returning the maximum damage is a conservative approach, which is the equivalent of saying that the highest fatigue damage experienced around the blade root for a given load history is experienced everywhere around the blade root. A more exact method would be to store the damage experienced at each location separately; however, because our testing indicated that the locations of maximum damage were all very close, it was appropriate to return a single maximum damage value.

## 3 Comparison of Fatigue Model to SOWFA/FAST Data

With our new model explained in Sec. 2, this section will show how it compares to the high- fidelity LES and loads simulations. All of the comparisons shown in this section demonstrate the damage a turbine experiences for different amounts of partial waking. In each scenario, there is one fixed upstream turbine, while a second downstream turbine is moved across the wake. The damage to the downstream turbine is shown for each location across the wake. Because of the computational expense required for the SOWFA and OpenFAST runs, the data points are more scarce than those for our new model. As was said

earlier, these results shown are for the NREL 5-MW reference turbine. There is a safety factor of 2.0 for the comparisons in this section (rounded up slightly higher than the 1.954 suggested in the blade definition (Resor, 2013)). Figures 10 and 11 show how our model compares to the high-fidelity SOWFA and OpenFAST data for a lower freestream turbulence-intensity case of 4.6%, and a higher freestream turbulence-intensity case of 8%. In these two figures, we show the damage results for wind speeds of 10, 11, 12, and 13 meters per second. These wind speeds are near rated speed, where the pitch angle is zero or very small, which means these wind speeds should experience the highest normal operation load fluctuations and associated fatigue damage. Remember that the damages shown in these figures is the total damage calculated by our model, which takes into account both the flatwise and edgewise loads.

Figure 10 shows how our model compares to the high-fidelity data for the low turbulence case. The model trends match the SOWFA and OpenFAST data very well, across all of the turbine spacings and wind speeds shown. Even the values of the model for most of the cases match the high-fidelity damage predictions, with the largest differences occurring for 13 m/s at 4 and 7 diameter turbine separation. At least part of the discrepancy between our model and the high-fidelity data at these wind speeds and separations can be explained with the wind speed model shown in Fig. 5. In this figure, notice that the model wind speeds do not match the SOWFA data as well, especially for a freestream velocity of 13 m/s at 4 and 7 rotor diameter separation. An additional cause for the difference between our damage results and the SOWFA/OpenFAST damage is the control scheme. The loads are sensitive to the blade pitch angle, so small differences in the controller or inflow data near rated speed will greatly affect the results. Another possible reason for the difference between our model and the SOWFA data is wind shear, which is better captured with the high-fidelity SOWFA simulation. Achieving such a good match to the damage values is particularly impressive because the final damage value is dependent on so many intermediate calculations that are required with high precision. It is expected that the model will not match the higher-fidelity data perfectly. The exact turbulence history from the SOWFA and OpenFAST data is difficult to match, and a large driver of the final damage value. Our model is predicting the turbulent effect on damage with a relatively small number of turbulence samples. Additionally, the simple fatigue model does not capture blade aeroelasticity or any dynamics of the system, which are considered in the higher-fidelity models.

Figure 11 shows how our model compares to the high-fidelity data for the high turbulence case. In this figure, we see similar trends to the low freestream turbulence results. The model predicts the trends very well and most of the actual damage values remarkably well for freestream wind speeds of 10 m/s and 11 m/s. For freestream wind speeds of 12 m/s and 13 m/s, for all rotor separation distances, our model greatly underpredicts the damage. To some extent, this difference can be explained by referring to Fig. 5. Especially at the wind speeds and separations where the model performs poorly, the wake model does not accurately predict the wind speed profiles. This could be improved with better velocity profiles, but part of the reason is simply that SOWFA captures physics that the analytic wake model does not, such as local speed up areas and wind speed gradients throughout the domain. For the wind speeds around rated power that we have examined in this paper, the slight differences in the wind speed estimation can have a large effect on the loads because this is the region where the blades start to pitch. Additionally, the accuracy of our fatigue model could likely be improved, perhaps significantly, with a better turbulence-intensity model. As shown in Fig. 8, the model predicts the turbulence-intensity profiles relatively well but certainly not perfectly. Although improving these intermediate models could improve the final fatigue damage calculations, this

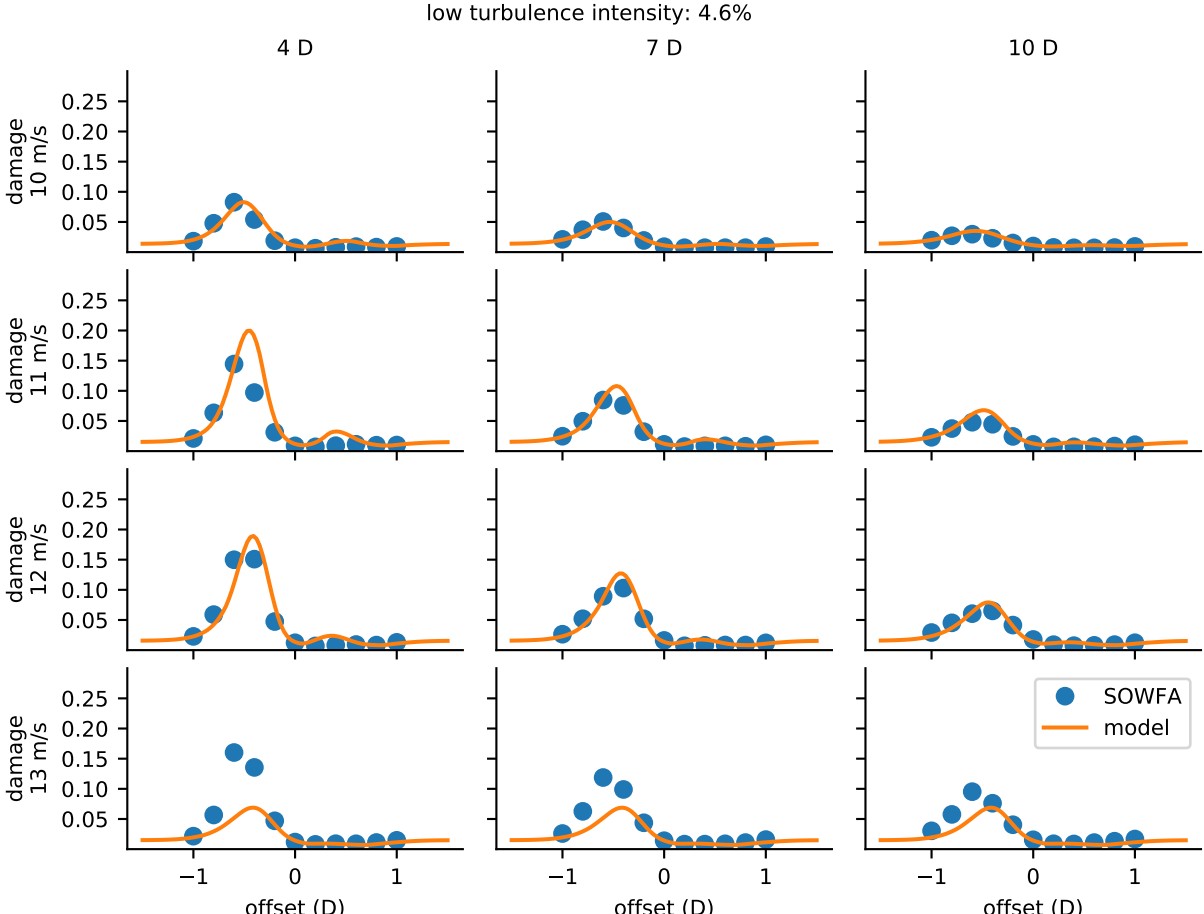

**Figure 10.** Comparison of our presented damage model to the lifetime damage predicted by the loads from the SOWFA and OpenFAST data. This figure is for a low freestream turbulence intensity of 4.6%. From top to bottom, each row represents different freestream wind speeds of 10, 11, 12, and 13 meters per second. From left to right, each column represents different distances from the upstream turbine of 4 D, 7 D, and 10 D.

is outside of the scope of this study. As is, the model predicts the locations of maximum damage very well. For optimization usage, we don't necessarily care about the exact damage value prediction. What is important is being able to identify areas where high damage occurs in order to avoid these places. We are very satisfied that our damage model does predict the actual damage value well for most inflow conditions; however, this is not as important as the locations of maximum damage.

One observation that is consistent across all of the results shown in Figs. 10 and 11 is that the fatigue damage is higher when the turbine is partially waked on one side (with a negative offset, in how we have presented the data), but the damage is slightly lower or at least unaffected when it is partially waked on the other side. While at first this may seem unintuitive, there is a simple explanation for this behavior caused by the interaction of the gravitational loads and the lead-lag aerodynamic loads.

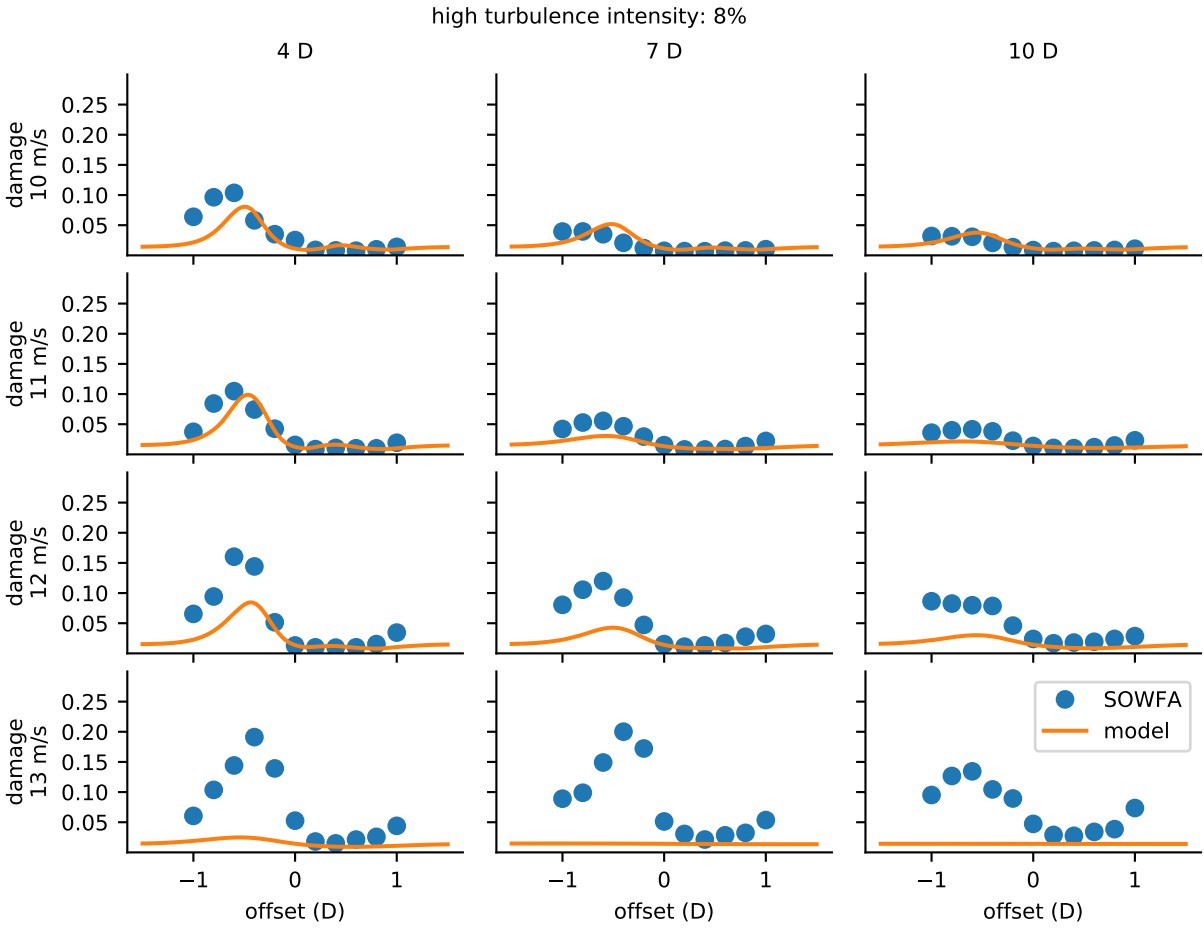

**Figure 11.** Comparison of our presented damage model to the lifetime damage predicted by the loads from the SOWFA and OpenFAST data. This figure is for a high freestream turbulence intensity of 8%. From top to bottom, each row represents different freestream wind speeds of 10, 11, 12, and 13 meters per second. From left to right, each column represents different distances from the upstream turbine of 4 D, 7 D, and 10 D.

If the blade is partially waked while rotating upward, the aerodynamic loads on the blade will be relatively lower. This means there is a smaller force to offset the gravitational loads, and the load fluctuations will be higher than if the turbine is operating in freestream conditions. On the other hand, if the blade is partially waked while rotating downward, the aerodynamic loads acting in the same direction as the gravitational force are relatively lower. In this configuration, the load fluctuations are smaller than in freestream operating conditions. These interactions are explained more clearly in Fig. 12.

## 4 Example Optimizations

In this section, we will discuss an example wind farm layout optimization in which we used our model to constrain the damage caused by partial waking throughout the wind farm. Before discussing the optimization and results, we'll briefly describe the models and assumptions we've made for this optimization. As with the rest of this paper, in the example optimization we assumed the NREL 5-MW reference turbine design throughout the farm. This is an upstream turbine which has a rotor diameter of 126.4 meters, a hub height of 90 meters, a cut-in wind speed of 3 meters per second, a rated wind speed of 11.4 meters per second, and a rated power of 5 megawatts. The power curve for this turbine was assumed to be perfectly cubic, as represented in Eq. 37.

$$P = \begin{cases} 0 & U_{\text{rotor}} < U_{\text{cut-in}} \\ (U_{\text{rotor}}/U_{\text{rated}})^3 P_{\text{rated}} & U_{\text{cut-in}} \leq U_{\text{rotor}} < U_{\text{rated}} \\ P_{\text{rated}} & U_{\text{rotor}} \geq U_{\text{rated}} \end{cases} \tag{37}$$

In this equation, $U_{\text{rotor}}$ is the inflow speed to the rotor, $U_{\text{cut-in}}$ is the cut-in wind speed, $U_{\text{rated}}$ is the rated wind speed, and $P_{\text{rated}}$ is the rated power. We assumed that tower shadow is negligible, meaning that the power is only a function of inflow wind speed with no adjustment required. Additionally, we assumed wind speeds that were all relatively low and thus did not need to consider a cut-out wind speed. We used a safety factor of 2.0 for the fatigue calculations and also assumed the turbines always faced directly into the oncoming wind and that the terrain was flat.

Figure 13 shows the wind rose and wind speed distributions we used for this optimization. The wind direction is mostly from north to south, or south to north, which could potentially lead to turbines that are partially waked a majority of the time. The wind speeds are close to the rated wind speed of the NREL 5-MW reference turbine. We used 100 wind direction bins (every 3.6 degrees) in each optimization. Because the turbine fatigue is worse for partially waked orientations, it requires a large number of wind direction samples. We found 100 bins to be sufficient to appropriately approximate the damage. We assumed directionally averaged wind speeds, meaning we assumed the wind speed from each direction was always the mean wind speed from that direction. We assumed a wind shear exponent of 0.15 and a freestream turbulence intensity of 4.6% (corresponding to Fig. 10 for the damage calculations). As discussed in Sec. 2.3, the wake model and local turbulence model we used for this paper used tuning constants to better represent the high-fidelity data. For simplicity, for the optimization results shown in this paper, we used a single set of tuning constants (regardless of the wind speed) of $c_1 = 0.00485$, $c_2 = 0.0115$, $\alpha = 2.66$, and $\beta = 0.542$ for the analytic wake model, and $C_1 = 1.18$ and $C_2 = 2.39$ for the local turbulence intensity model.

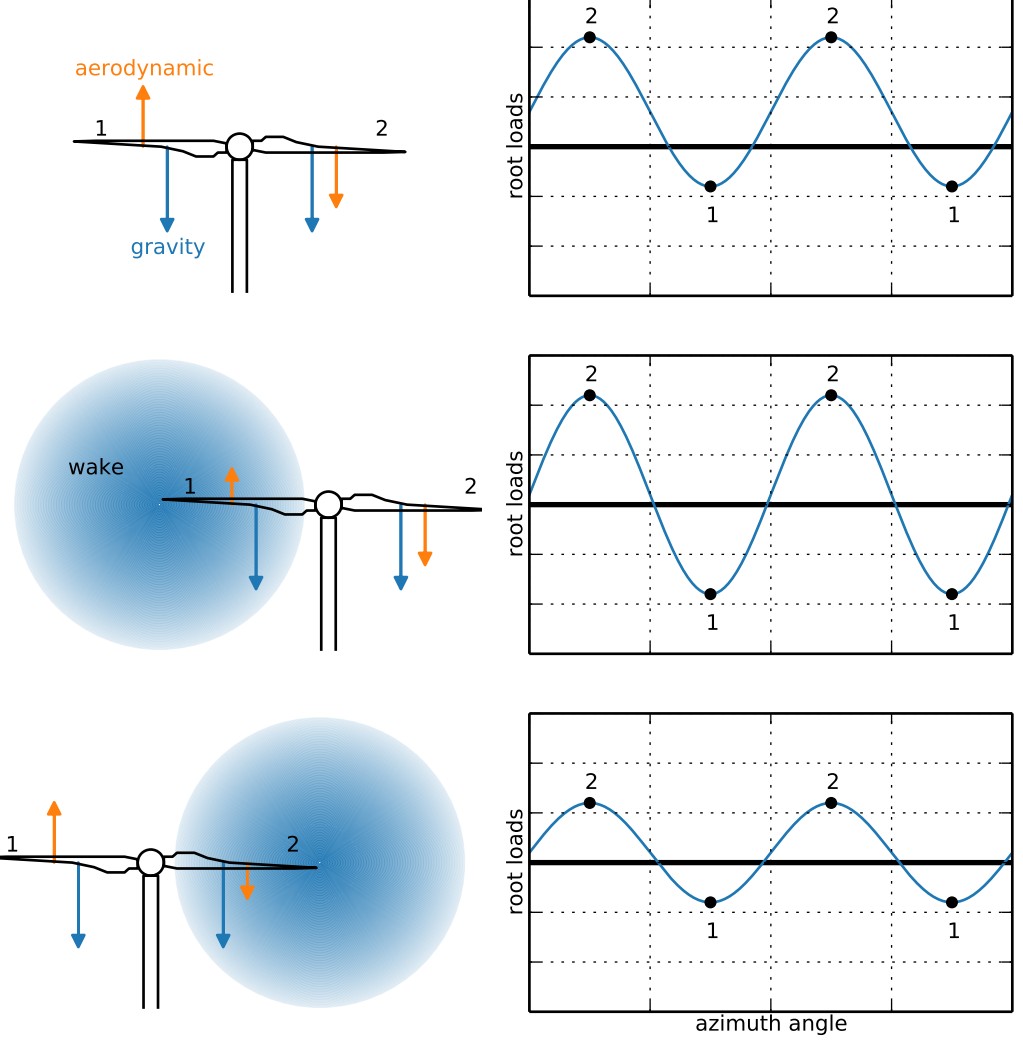

**Figure 12.** Exaggerated lead-lag load differences for different waking scenarios. Shown conditions are freestream, partially waked that increase load fluctuations, and partially waked that decrease load fluctuations. The different combinations of the gravitational force and the aerodynamic force along the blade cause different load fluctuations. Blade positions 1 and 2 are labeled on the turbine figures on the left, which correspond to the numbered points in the loading figures on the right.

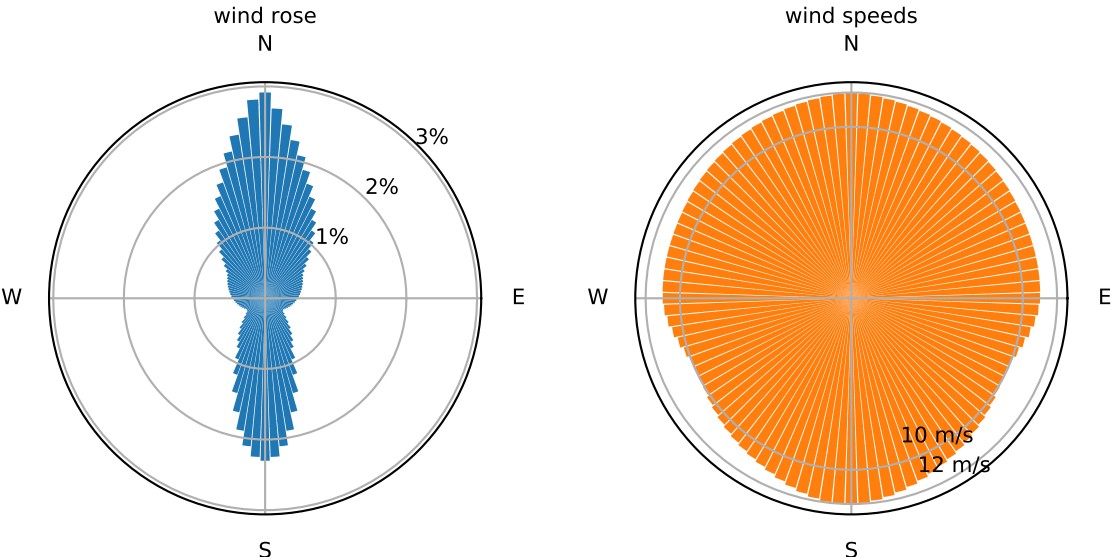

**Figure 13.** The wind rose and directional wind speeds used in our optimization. The data are divided into 100 bins, one every 3.6 degrees.

The objective of this optimization was to maximize the annual energy production (AEP) of the wind farm with respect to the location of each turbine. The rotor hubs were constrained to be at least two D apart from each other. Additionally, there were boundary constraints that forced the turbines to remain in a fixed wind farm boundary. Finally, the total fatigue damage was constrained to be less than a desired value. This fatigue damage was calculated with our model presented in this paper, which takes into account both the flatwise and edgewise loads. When considering fatigue damage, the assumption is that failure occurs when the damage value reaches unity. However, with the layout optimization, we are only considering the additional damage caused by waking and partial waking of the wind turbines. Other significant drivers of fatigue are extreme wind gusts and cases of extreme wind shear and veer. These phenomena are not captured with our presented model; therefore, we must constrain the optimization to some value less than one. In full, this optimization is represented in Eq. 38.

$$
\begin{array}{ll}
\text{maximize} & \text{AEP} \\
\text{w.r.t.} & x_j,\ y_j\ (j = 1, \ldots, \text{nTurbs}) \\
\text{subject to} & \text{boundary constraints} \\
& \text{spacing constraints} \\
& \text{damage}_j < \text{damage}_{\text{max}}\ (j = 1, \ldots, \text{nTurbs})
\end{array}
\tag{38}
$$

We used the optimizer SNOPT for this example, which is a gradient-based optimizer that works well for large problems with many design variables and constraints (Gill et al., 2005). We provided exact, analytic gradients to the optimizer using ForwardDiff, an automatic differentiation package in Julia (Revels et al., 2016). In order to provide a point of reference and find a starting point for our final optimization with damage constraints, we first ran 50 optimizations of the wind farm without

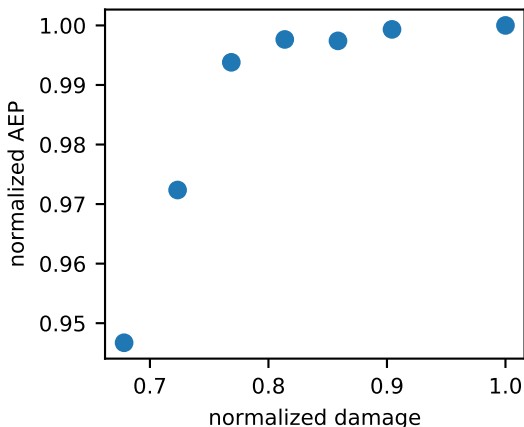

**Figure 14.** Pareto front displaying the trade-offs between maximizing AEP and minimizing turbine damage. Both axes have been normalized by the AEP and maximum turbine damage achieved without constraining the turbine damage.

damage constraints and with randomly initialized design variables. We then chose the layout with the highest AEP and used that as the starting point for our optimizations with loads constraints.

## 4.1 Multi-Objective Optimization

The example optimization we performed was for a wind farm with 40 wind turbines and with a parallelogram-shaped boundary. The average turbine spacing for this wind farm was 4 D, meaning that the turbines densely populated the wind farm. Without damage constraints, the wind farm had a maximum turbine damage value slightly above 0.055. The exact value for the damage constraints that we applied was relatively arbitrary; there was no special significance to the values that we chose (discussed below). The actual damage values were very sensitive to the factor of safety and the ultimate stress used in the calculations. Therefore, rather than pay much attention to the exact values, we normalized the results in order to demonstrate the relative reduction in damage that can be achieved.

We performed a multi-objective optimization in order to study the trade-offs between optimizing for power production and minimized loads. We used the epsilon-constraint method to create a Pareto front of the maximum AEP and maximum turbine damage for the example optimization. Multi-objective optimization using the epsilon-constraint method is performed by repeatedly optimizing an objective, with successively stricter constraints. In our specific case this meant to repeatedly maximize the AEP, while successively decreasing the maximum allowable turbine damage. Figure 14 shows the Pareto front from this optimization. In this figure, the axes have been normalized by the AEP and maximum damage obtained when the layout was optimized without any damage constraints.

There are two main observations that we can take away from this Pareto front. First, there was a large flat area in the Pareto front, which indicates that we can constrain the maximum damage by a significant amount with very little impact to the AEP.

The damage can be reduced by almost 10% for an AEP reduction of only 0.07%. Additionally, the damage can be reduced by more than 20%, with a very small AEP reduction of 0.6%. The damage can be significantly constrained practically for free. After that, there was a steeper drop off in the AEP for additional reduction of maximum damage. Second, the largest damage reduction shown can be achieved with a small impact on the AEP. A 5% reduction in AEP is certainly not negligible; however, it is noteworthy that a more than 30% reduction in maximum turbine damage can be applied with such a small reduction in AEP. As has been discussed in detail, partial waking can greatly increase the fatigue damage on a wind turbine. In addition to increasing damage, partial waking is also detrimental from a power production perspective because any amount of waking decreases the inflow wind speed to a turbine and, therefore, the power produced by the turbine. An optimal wind farm layout that maximizes AEP will aim to reduce any waking as much as possible. This means that a wind farm optimized to maximize energy production, and a wind farm optimized to minimize turbine damage, share many similarities. Each objective can achieve a desirable value without forcing a significant trade-off from the other.

### 4.2    The Effect of Damage Constraints on the Optimal Solution

With the optimal results from the multi-objective optimization shown in the previous section, we will now explore the differences between the optimal solutions in greater detail. We will compare the optimal layouts and turbine damages between the layout optimized with no damage constraints and the layout with the most strict damage constraints, corresponding to the point on the top right and the point on the bottom left in Fig. 14, respectively.

Figures 15, 16, and 17 show these comparisons. Figure 15 shows the optimal turbine layouts with and without damage constraints. The AEP for the turbine layout without damage constraints is 1,129 GWh, while the AEP for the layout with damage constraints is 1,069 GWh. With the optimization method we used, there is a sacrifice in AEP of 5.3% required to meet the damage constraints. This AEP loss is quite significant; however, it is impressive to realize that the damage from partial waking can be reduced by the much larger value of 32%. Notice the differences in the layouts with and without damage constraints.

Because of the complex interactions that happen between all of the turbines in the wind farm, and the many wind directions considered in the farm analysis, it is difficult to draw many conclusions simply by looking at the wind farm layouts. However, looking closer at the optimal layout with damage constraints, there are two observations we can make. First, the overall layout still appears relatively similar to the optimal layout without damage constraints. The objective of the optimization was still to maximize AEP. The wind rose for this optimization was predominately from the north and south, which indicates that the turbines would produce more energy if the distance between them was maximized for these dominant wind directions. Second, the turbines that are close together and would cause partial waking when the wind is coming from the north and south are partially waked on the side that does not greatly increase fatigue damage. We circled these turbines in Fig. 16. In Figs. 10 and 11, we can see that when partially waked on one side, the damage greatly increases, especially when the turbines are close together. In Fig. 16, we notice that there are many turbines that are close together, but when they cause partial waking from the dominant wind directions, they are partially waked on the better side for fatigue damage.

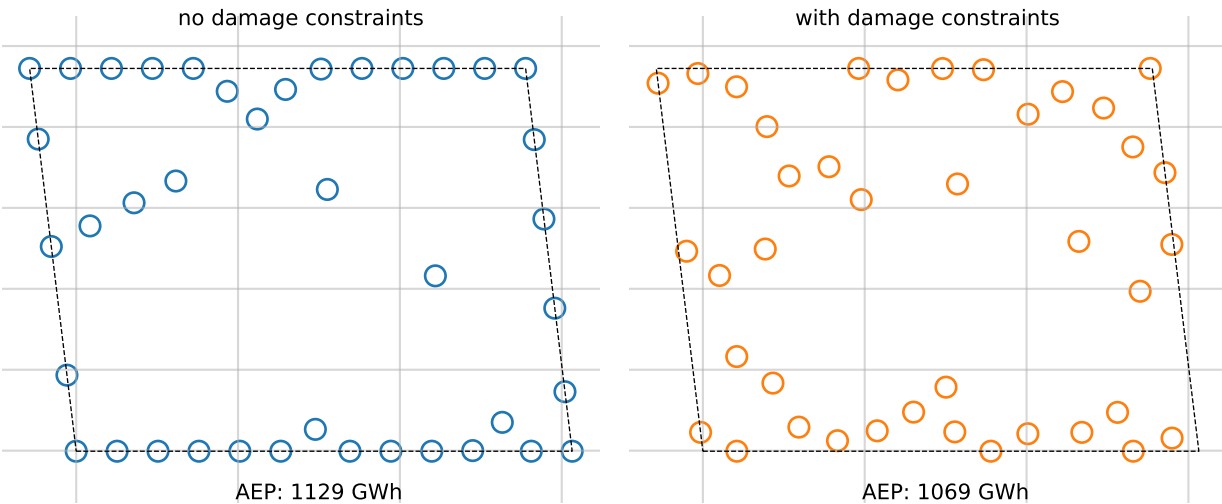

**Figure 15.** The optimal layout results from our optimization. On the left are the optimal turbine locations when the turbine layout is optimized without damage constraints. On the right are the turbine locations when damage from partial waking is constrained to be less than 0.04. The dotted black lines represent the wind farm boundary and the circles represent the wind turbines, with the circle diameter accurately scaled to represent the turbine rotor diameter.

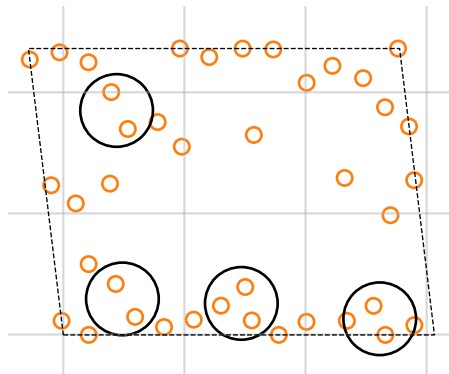

**Figure 16.** The layout optimized with damage constraints, with the turbine pairs that would cause partial waking from the dominant wind directions circled.

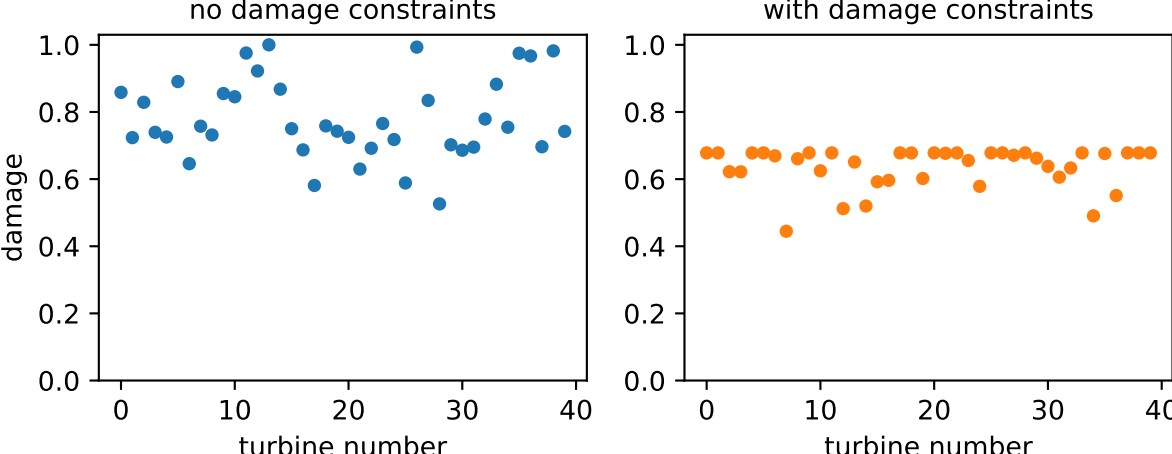

**Figure 17.** The fatigue damage of each turbine in our wind farm layout optimizations. On the left are the damages for the wind farm layout optimized without damage constraints, while on the right are the damages with the maximum damaged constrained to be less than 0.04. These plots have been normalized by the maximum turbine damage in the layout optimized with no damage constraints. On the right, the line of maximum damages around 0.7 are the normalized damage constraint of 0.04.

Figure 17 shows the total damage accumulated by every turbine for each of the layouts shown in Fig. 15. These damages are normalized by the maximum turbine damage for the layout optimization without loads constraints. When the layout was optimized without damage constraints, there was a large spread of damage values experienced by the different turbines. More than half of these values are above the desired damage constraint of 0.04 ($\approx 0.7$ when normalized). With damage constraints activated, we were able to reduce the damage from partial waking to the desired value for every turbine in the wind farm. Additionally, the spread of turbine damages was greatly reduced. Also notice how many turbines are right on the damage constraint value. For about half of the turbines in the wind farm, the damage constraint was active.

## 5   Conclusions

In this paper, we presented a model to predict fatigue damage on wind turbines caused by partial waking throughout a wind farm. The model predicts the trends of turbine damage very well, including the partial waking that results in the worst damage. The model we presented can be used in an optimization framework to constrain or minimize the turbine damage throughout the wind farm. We demonstrated using the model in an example in which we optimized the layout of turbines in a wind farm while constraining the damage caused by partial waking. We found that, at least in the case that we explored, the damage could be successfully reduced by 30% with a sacrifice to the optimal AEP of around 5%. Additionally, we performed a multi-objective optimization, which shows that when we reduced the maximum turbine damage by 10%, the sacrifice to AEP was only 0.07%, and when we reduced the maximum turbine damage by 20%, the AEP sacrifice was only 0.6%. These results were for a wind

farm where the turbines are spaced close together and a unidirectional wind rose. We expect that wind farms where the turbines are farther apart and more distributed wind roses could achieve large damage reductions for smaller sacrifices to the AEP.

The area of loads and fatigue consideration in wind farm layout optimization has huge potential for continued research. For continuation of this research paper, we have a few specific recommendations. First, further validate and improve our proposed damage model with more SOWFA runs for a wide variety of wind conditions. In this paper we have presented a range of wind speeds, amounts partial waking, distances downstream, and two ambient turbulence intensities. Further confidence could be achieved with more high fidelity data. In addition to more SOWFA runs, it would also be insightful to compare our model to actual wind farm data. Although real wind farm data would not provide all of the cases that we would want to consider, they would be important points to validate the model. Second, conduct a deeper investigation into how including damage constraints in wind farm layout optimization affects farm design and performance. For the example optimization we have shown, and in simple optimizations we have run in the past (Stanley et al., 2020), we found that the damage can be constrained with a small sacrifice to the energy production. Considering a wider variety of wind farms and wind resources would be useful to confirm or clarify this conclusion. Thrid, couple our proposed model with active wind farm control optimization. Additional damage reductions are likely achievable by coupling layout and control optimization. Using active yaw control for wake steering would be particularly interesting with fatigue damage constraints. As we demonstrated in this paper, the partial waking is detrimental for fatigue damage on one side of the turbine and negligible or significantly lower on the other side. Wake steering through yaw control gives the freedom to deflect wakes in either direction behind a wind turbine, which could increase power production by partially waking downstream turbines on the side that does not increase fatigue damage. Fourth, optimize a wind farm layout in which the turbine damage is part of the objective. This could mean optimizing the lifetime energy production of a wind farm in which the turbines would produce energy for more years if the damage they experience from partial waking is reduced. Fifth, investigate the sensitivities and uncertainties involved with each of the models and assumptions made throughout the model, and how they impact the final damage calculations. This would be incredibly relevant for future studies that specifically include uncertainty analysis. The method presented in this paper uses analytic models, but we expect that the final results are sensitive to model parameters, tuning variables, and uncertainty in any inputs. A better understanding of these uncertainties would be important in building reliable wind farms.

Recent years have seen significant improvements in wind energy technology and large reductions in the cost to produce wind energy. The model that we have proposed is an additional improvement, allowing wind farm layout optimization that considers turbine loading and fatigue, which will further decrease the cost of wind energy.

**Code and data availability**

With the exception of the optimizer SNOPT, all of the code and models used in this study are open source. The wind farm models, optimization framework, and damage model are available here: https://github.com/byuflowlab/FlowFarm.jl. The individual run scripts and plot generation scripts are available in the journal branch here: https://github.com/pjstanle/loads-journal.

## Author Contributions

APJS led this research, including developing and testing potential fatigue methods, running the optimizations, and writing the paper. JK set up and ran SOWFA to obtain the high-fidelity flow field data and helped develop ideas. CB ran OpenFAST with the high-fidelity flow field data and helped develop ideas. AN helped develop ideas and methodology, provided feedback throughout the entire process, and provided editing for the paper.

## Competing Interests

The authors declare no competing interests.

## Disclaimer

The views expressed in the article do not necessarily represent the views of the DOE or the U.S. Government.

## Financial Support

Funding provided by the U.S. Department of Energy Office of Energy Efficiency and Renewable Energy Wind Energy Technologies Office.

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

## Appendix A: Large Eddy Simulation

We used Simulator fOr Wind Farm Applications (SOWFA) to generate the inflow data which we then used to calculate the loads with which to compare our model. SOWFA is a high-fidelity large eddy simulation tool that was developed at the National Renewable Energy Laboratory for wind farm studies. It is based on the open source CFD solver OpenFOAM, and can be coupled with NREL's FAST modeling tool. SOWFA has been used in several previous wind farm control studies (Fleming et al., 2013, 2015; Gebraad et al., 2016).

In this paper, SOWFA uses an actutator disk model to represent the turbine in an atmospheric boundary layer. It solves the three-dimensional incompressible Navier-Stokes equations and transport of potential temperature equations, which take into account thermal buoyancy and Earth rotation (Coriolis) effects in the atmosphere. The inflow conditions for this simulation are generated using a periodic atmospheric boundary layer precursor with no turbines. Additional details can be found in (Fleming et al., 2013).

All simulations performed in this study used a neutral boundary layer, a shear exponent of 0.12, and were simulated in a 5-km$\times$2-km$\times$1-km domain. Low turbulence cases had approximately 4.6% turbulence intensity, and high turbulence cases had approximately 8% turbulence intensity. Simulations were run for the following cases:

- 10 m/s low and high turbulence

- 11 m/s low and high turbulence

- 12 m/s low and high turbulence

- 13 m/s low and high turbulence

Inflow data for the OpenFAST simulations were generated based on the respective SOWFA simulations for the different cases. Slices of the SOWFA data were taken at different distances downstream from an upstream turbine, as shown in Fig. A1.

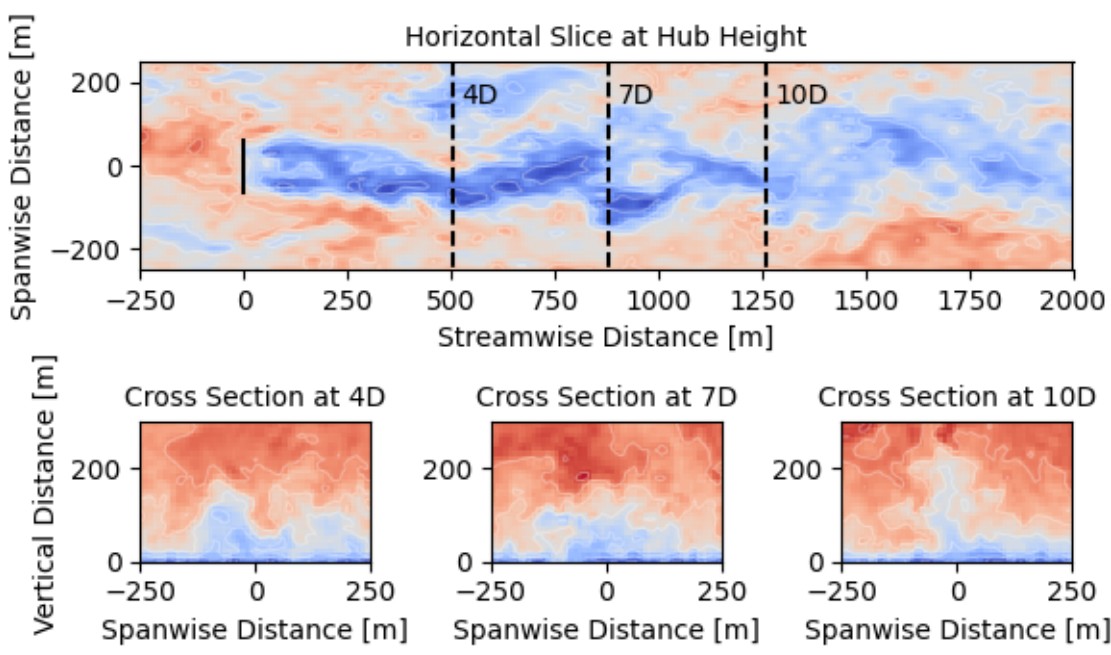

**Figure A1.** Examples of SOWFA data used to generate inflow for FAST simulations.