# Peer review of "A Model to Calculate Fatigue Damage Caused by Partial Waking during Wind Farm Optimization"

_Wind Energy Science, 2020_

## Referee Comment (RC1) · Wim Bierbooms (Referee) · 29 Jan 2021

The manuscript deals with an important issue with respect to wind farm optimization. The proposed inclusion of fatigue damage will certainly be applied soon in practice.

Major Remarks

- the applied coordinate systems as well as the different wind speeds (undisturbed, wake, instantaneous, averaged over blade, averaged over rotor) should be appropriate introduced, e.g. in an added section 2.0; use can be made of e.g. graph of a wind turbine like fig. 7. Furthermore, all names / symbols should be consistently used. The term "effective" should be avoided since it is unclear. Furthermore a capital letter should be used for an averaged value and a lower case letter for instantaneous values), e.g. mean wind speed in wake averaged over rotor: $U_{wake\text{-}rotor}$ ; instantaneous wind speed averaged over blade: $u_{blade}(\psi,t)$
- In line with the point above it would be better to indicate the velocities and TI in step 3 and 4 with wake velocities and wake TI. Furthermore, it would be better to move lines 247-252 to step 5 (averaging over a blade) and line 253-254 to step 8 (considering the entire rotor)
- The weakest point of the proposed method is the generation of turbulence samples, section 2.1. It can easily be improved by e.g. using the method of Veers
https://prod-ng.sandia.gov/techlib-noauth/access-control.cgi/1988/880152.pdf

  By doing so the turbulence will have the correct spectrum (which is essential for a fatigue analysis). Since in section 2.7 the instantaneous wind speed is needed, varying with azimuth, the so-called rotational sampled wind speed should be used, e.g. at a radius of ¾ R. This can be obtained by first generating a wind field on a rectangular grid (Veers) and next take the wind speed as seen by a blade element (at ¾ R) rotating through this turbulent wind field.

- Line 128 / 129: "The tuning constants … depend on the blade azimuth angle"; it is unclear why that is the case since there is no periodic loading (like yaw, shear, turbulence and tower shadow)
- Line 272: "just considering these two azimuth angles is sufficient"; this implies that each revolution will lead to exactly 1 load cycle. In reality, each revolution will contain plenty of smaller load cycles as well. Since fatigue behaves rather nonlinear one can't tell in advance that neglecting these smaller load cycles is allowed.
- Comparison with SOWFA/FAST; an appendix can be added in which the steady turbine response is compared (i.e.: $C_T$, P, $\Omega$, $\theta$, $M_{flatwise}$, $M_{edgewise}$) for just 1 wind speed (say 13 m/s) during say 4 to 5 revolutions. In the model (section 2) the same wind input should be used

Minor Remarks

- Line 27: add "wind shear"
- Line 77: mention one of these "interactions"
- Line 95: change "wind speed" into "undisturbed mean wind speed";  see also 1[st] bullet point Major Remarks
- Line 99/100: change "effective wind speed across the blade" into "wind speed averaged over blade";  see also 1st bullet point Major Remarks
- Section 2.2: since the Loads Surrogates are derived only once, it is not clear why not the more sophisticated package FAST has been used

- Eq. (2); perhaps it is better to use the symbol q for a force per length instead of F
- Table 1: mention the units of the constants a to g as well as Psi and Theta
- Figure 5: why does the x-axis not continue until the cut out wind speed? Mention if these curves apply for the case Theta is less then OR greater than 0.05 rad. (Table 1)
- Line 141: This is in contradiction to the outcome of a BEM calculation I did (based on the NREL 5 turbine): the effect of a change of 1 degree in pitch correspond to about a variation of rotational speed of 8 %
- Eq. 5: Delta_u and u_infinity: use capital letters instead; change "d" into "D"
- Line 153: also introduce delta and z_h
- Line 168: add: and delta=0
- Eq. (10): add nTurbs (upper bound summation)
- Figure 7: add a figure with the variation of the wind speed as function of azimuth (for an offset of e.g. 1D) and compare with the averaged value (over the azimuth) as well as the average of the 4 sample points
- Figure 8; "offset" has not been properly introduced yet
- Title section 2.4: change "Intensity" into "Intensities" (in line with Fig. 1)
- Line 205: change "mean" into "undisturbed mean"
- Page 13: Also introduce delta (from Eq. (11))
- Eq. (23): add, for clarity: TI(r)=TI_a + Delta_TI, with Delta_TI given by Eq. (11)
- Title 2.5: adopt; see 1st bullet point Major Remarks
- Eq. (24): change U into U(r); + mention the equation for U(r) (is it Eq. (10)?)
- Section 2.6; show in an appendix a few examples of wind speed, rotational speed, pitch angle and bending moments varying over the azimuth angle.
- Eq. (26): adopt; see 1st bullet point Major Remarks; make 2 versions (Eq. 26a and 26b): one with TI averaged over blade (from Eq. (23)) and one with TI averaged over rotor
- Line 282: I guess it should be step 4 (instead of 3)
- Section 2.9 / line 296: refer to Eq. (26a)
- Section 2.7: refer to Eq. (26b)
- Line 361: change step 7 into step 8
- Line 404 (end): typo "us"
- Line 442: add a statement about tower shadow
- Line 495: skip "about 6 times more" (such a comparison doesn't make sense)
- Caption Figure 17: add for clarity that 0.04 corresponds with 0.07 normalized
- Line 521: skip digit: about 5%
- Section 5: you may add for further research: to performan several SOWFA simulation in order to determine the spread of the SOWFA results (Fig. 6, 9, 10 and 11)
- Line 631: is wind shear included?

---

## Referee Comment (RC2) · Anonymous Referee #2 · 19 May 2021

**Comments on "A Model to Calculate Fatigue Damage Caused by Partial Waking during Wind Farm Optimization"**

**General comments:**

The authors present an interesting approach to make efficient fatigue estimation for wind farms viable; in particular for use in optimization-based design approaches or possibly in future systems for operation and maintenance. By utilizing analytical models and empirical surrogate models, the study provides a simplified, but transparent methodology for estimating the blade fatigue and shows how this can be used in a simplified optimization context to make decisions about the wind farm layout. A few of the subsections could use additional details and explanations and, as will be explained below, there is a common theme of error estimation that could be considered and/or underlined to strengthen the paper.

**Specific comments:**

What exactly is shown in Fig 2? The y-axis label is not clear and hence it is not clear what is even measured on the y-axis. The caption indicates that this is an "example set" of turbulence samples, but the text on the same page indicates that the figure shows exactly the samples used in the paper. Which is it?

Any further comments on the Gaussian Wake model (Page 8-11)? It is indicated that the authors "found good results" with it, presumably when comparing with the SOWFA data (?). Are there any effects that this model is expected to miss? Do you have error estimates for the tuning constants in Tab 1? It would be instructive to include these in the table (as +/-) or show the overall effect on the surrogate fit by having error bars in Fig 4 and 5. If the errors are very small, a short comment to this effect in the text would suffice.

Page 11, eq 10 and below: The authors provide a reference to justify the use of the linear wake summation model, but a few more comments here would be instructive (whether the content of these can be found in the reference or not). E.g. what is the motivation for using the linear model over others besides the fact that it "works well with the Gaussian wake model"? Is it used for superior accuracy alone or is it more a case of a simpler model that works acceptably well without introducing further complications? Is there any downside to using this model?

Similarly as above, any further comments on the turbulence intensity model (Page 12-14) chosen and any possible impact of this choice? Any error estimates for the

tuning constants in Tab 3 (or possibly the overall effect on the error level of the model shown in Fig 9)?

What is the expected error level of the surrogate model described in Section 2.9 on Page 17?

It might be instructive to illustrate a bit more clearly how a load/moment "history" is obtained via the Turbulence and Azimuth Loop, perhaps through some example. Specifically, how this method produces something analogous to the conventional load time series obtained from simulations that are usually the input to rainflow counting-based fatigue assessment methods.

It is indicated on Page 19 (line 360) that the results in the paper are based on load histories obtained from 50 complete revolutions of the rotor. For the NREL 5MW this would be something like a few minutes of simulation time. Conventional time domain-based fatigue estimates are usually based on at least 60 minutes of simulation. The shorter duration is understandable for the purposes of the paper, but do the authors have any comment on this?

Fig 10 and 11: The y-axis labels should indicate that the values shown are in fact lifetime fatigue values (which is my assumption, but this is not clear).

In the Conclusion, on Page 30, the use of active yaw control and its possible coupling to the proposed method is discussed for future research. As noted previously in the paper, the yaw angle was fixed at zero in this study. Any comment on what effect (if any) non-zero yaw angles (or yaw errors even) might have on the proposed method?

I have made several comments concerning error estimates for various parts of the proposed method. Beyond the general interest of such error estimates as indicators for the validity of each simplification, the analytical nature of the authors' methodology actually makes it possible to potentially propagate these errors all the way to the end fatigue result. The resulting error estimates could be very useful and would in fact be a strength of the method. In particular, for optimization it could provide some manner of error bound or expected uncertainty in the result that would show the level of robustness of the solution. Especially in light of Fig 14 and related results. It could also make optimization approaches that consider uncertainty more explicitly more viable for use in similar wind farm studies. While any larger investigation into this issue or indeed carrying out such an error propagation might be out of the scope of the present work, some discussion of these points would be favorable for the paper.

Do the authors have any comments regarding the use of SOWFA as a benchmark for the accuracy of the proposed method and to what extent SOWFA itself can be used in this manner (i.e. its accuracy)? It is likely worth pointing out that any relevant experimental data for windfarms, which may not be available at present, could presumably be similarly used to tune the parameters of the method, so it is not reliant on SOWFA as such.

---

## Author Response (AR1)

**Response to Reviewers**

Andrew P. J. Stanley, Jennifer King, Christopher J. Bay, and Andrew Ning

**Reviewer 1**

First, we would like the express our gratitude for your review of our paper. We realize that you took time out of a busy schedule to read this manuscript and provide feedback, for which we are very grateful. We have structured this response to be clear and easy to follow. Each of your original comments will be shown in blue, immediately followed by our response in black. Note that the page number/line number indicating where the minor remarks are referring to is for the original submission. The revised manuscript will be slightly offset as we add/edit content.

The manuscript deals with an important issue with respect to wind farm optimization. The proposed inclusion of fatigue damage will certainly be applied soon in practice.

**Major Remarks**

the applied coordinate systems as well as the different wind speeds (undisturbed, wake, instantaneous, averaged over blade, averaged over rotor) should be appropriate introduced, e.g. in an added section 2.0; use can be made of e.g. graph of a wind turbine like fig. 7. Furthermore, all names / symbols should be consistently used. The term "effective" should be avoided since it is unclear. Furthermore a capital letter should be used for an averaged value and a lower case letter for instantaneous values), e.g. mean wind speed in wake averaged over rotor: Uwake-rotor ; instantaneous wind speed averaged over blade: ublade($\psi$,t)

We agree, this can be much more clear in the paper! We have made the following changes in the revised manuscript:

1. We added a paragraph in the beginning of section two to appropriately introduce the different wind speeds and turbulence intensities that apply to a point value, average over the entire rotor, or average over a single blade.

2. We removed all uses of "effective" referring to the wind speed or the turbulence intensity. This was unclear, and reworked to clarify that it was referring to the average wind speed acting over the blade or the entire rotor.

3. We changed the instantaneous variable of wind speed to be a lower-case $u$.

We did not add an additional figure, we feel like the current descriptions and equations are sufficient to understand each variable, and what an additional figure would add to the reader's understanding is minimal.

In line with the point above it would be better to indicate the velocities and TI in step 3 and 4 with wake velocities and wake TI. Furthermore, it would be better to move lines 247-252 to step 5 (averaging over a blade) and line 253-254 to step 8 (considering the entire rotor)

For clarification of the velocities and TI, we certainly agree. We believe the changes we made for the previous comment should clarify this as well. In regards to moving the explanations of the TI calculations to subsequent sections, we definitely understand the logic behind this move. For some it may be more clear to have this organization. However, we think the current organization which discusses the wind speed and TI calculations in separate sections is also acceptable and clear. Additionally, our current organization makes it more clear that the TI calculations do not need to be made within the turbulence and azimuth loop.

The weakest point of the proposed method is the generation of turbulence samples, section 2.1. It can easily be improved by e.g. using the method of Veers https://prod-ng.sandia.gov/techlib-noauth/access-control.cgi/1988/880152.pdf By doing so the turbulence will have the correct spectrum (which is essential for a fatigue analysis). Since in section 2.7 the instantaneous wind speed is needed, varying with azimuth, the so-called rotational sampled wind speed should be used, e.g. at a radius of 3/4 R. This can be obtained by first generating a wind field on a rectangular grid (Veers) and next take the wind speed as seen by a blade element (at 3/4 R) rotating through this turbulent wind field.

This is a very helpful comment. We have added to the explanation of the turbulence samples the following sentences to address this comment and orient readers.

"Although the turbulence samples used in this paper have the important statistical qualities required for the fatigue calculations in this paper, there are a variety of other methods that could be used to generate the turbulence values. One method could be to use the Sandia method, also known as the Veers method, introduced in 1988 (cite). Another could be to us the turbulence generator TurbSim to generate the turbulence samples, which has made several improvements since the Sandia method was introduced (cite). Using one of these methods could create more realistic turbulence history, but requires using an external program. For the results shown in this paper, the turbulence samples we generated are sufficient for demonstrating our method, and had appropriate statistical properties to compare well with high fidelity simulations."
* * *
Line 128 / 129: "The tuning constants ... depend on the blade azimuth angle"; it is unclear why that is the case since there is no periodic loading (like yaw, shear, turbulence and tower shadow)

The tuning constants vary slightly based on azimuth angle because the turbine that we used had a non-zero tilt angle. This was clarified in the text.
* * *
Line 272: "just considering these two azimuth angles is sufficient"; this implies that each revolution will lead to exactly 1 load cycle. In reality, each revolution will contain plenty of smaller load cycles as well. Since fatigue behaves rather nonlinear one can't tell in advance that neglecting these smaller load cycles is allowed.

In addition to the explanation given throughout the rest of this paragraph, we changed the last sentence to the following to acknowledge the presence of small load fluctuations:

"In reality, there are a multitude of small fluctuations that occur throughout the entire rotor rotation. However, for most conditions, just considering these two azimuth angles is sufficient as they capture the largest load differences which contribute the most to fatigue damage."
* * *
Comparison with SOWFA/FAST; an appendix can be added in which the steady turbine response is compared (i.e.: CT, P, $\Omega$, $\theta$, Mflatwise, Medgewise) for just 1 wind speed (say 13 m/s) during say 4 to 5 revolutions. In the model (section 2) the same wind input should be used

Great idea. This would certainly be an interesting figure to demonstrate how the higher fidelity SOWFA/FAST modeling compares with the steady-state CCBlade. Like many other potential figures we agree this would be interesting, however the not quite appropriate for this paper. Especially when we consider how lengthy the initial manuscript is, plus the additions we have made to address the excellent feedback we have received, we feel that we can rely on previous documentation and publications of BEM methodology.

**Minor Remarks**
* * *
Line 27: add "wind shear"

This was added to the revised manuscript.
* * *
Line 77: mention one of these "interactions"

This was added to the revised manuscript.
* * *
Line 95: change "wind speed" into "undisturbed mean wind speed"; see also 1st bullet point Major Remarks

This change was incorporated into the revised manuscript.
* * *
Line 99/100: change "effective wind speed across the blade" into "wind speed averaged over blade"; see also 1st bullet point Major Remarks

This phrase was clarified in line with the 1st bullet point of the Major Remarks.
* * *
Section 2.2: since the Loads Surrogates are derived only once, it is not clear why not the more sophisticated package FAST has been used

Good question. In our original formulation, there was no surrogate, but the loads were calculated directly inside the optimization using CCBlade. We added the following sentence to address your comment:

"A higher fidelity model could also be used to calculate the loads for this step, and our choice to use CCBlade was to allow for an easy transition to evaluating the loads directly in the optimization loop if desired."
* * *
Eq. (2); perhaps it is better to use the symbol q for a force per length instead of F

This was changed in the updated manuscript:

$$M = \int_0^{R_{\mathrm{tip}}} q(r) r \ dr$$
* * *
Table 1: mention the units of the constants a to g as well as Psi and Theta

Units were added for a to g in the table, and Psi and Theta units were added in the table caption.
* * *
Figure 5: why does the x-axis not continue until the cut out wind speed?

Thanks for this comment! We extended the x-axis out to the cut out wind speed of 25 m/s. Even though the wind speeds we used in this paper never reached that high of a value, we agree that it is important to include in this figure. In addition, we removed the figure showing the surrogate fit with no pitch. This was unnecessary because only the surrogate with pitch was used in the fatigue calculations.
* * *
Mention if these curves apply for the case Theta is less then OR greater than 0.05 rad. (Table 1)

We're not exactly sure what you mean with this comment, but will do our best to respond here. The blade pitch is determined by the average turbine wind speed (shown in Fig. 3). The surrogate constants that are used from Table 1 are then determined by the pitch angle of the blades. The figure showing the comparison of the surrogate to the higher fidelity data contains calculations that were made with both constant values, depending on the inflow wind speed/blade pitch.

Line 141: This is in contradiction to the outcome of a BEM calculation I did (based on the NREL 5 turbine): the effect of a change of 1 degree in pitch correspond to about a variation of rotational speed of 8 %

Yes excellent point. The variation of pitch angle may very well affect the rotation speed of the turbine. The statement on line 141 is referring to the sensitivity of the *loads* to the pitch angle and rotation speed. The loads are sensitive to the pitch angle, but not very sensitive to the rotation speed. This does not mean that rotation speed will not vary with pitch.

Eq. 5: Delta_u and u_infinity: use capital letters instead; change "d" into "D"

Apologies, we're not certain what you mean with this comment, as the delta before the u is already capitalized. Perhaps you are suggesting to capitalize the $u$'s in this equation, which we think is a good idea for consistency. We have gone through and changed all lower case $u$'s in equations to upper case.

Line 153: also introduce delta and z_h

These variables were defined.

Line 168: add: and delta=0

This was added.

Eq. (10): add nTurbs (upper bound summation)

This was added to the updated manuscript.

Figure 7: add a figure with the variation of the wind speed as function of azimuth (for an offset of e.g. 1D) and compare with the averaged value (over the azimuth) as well as the average of the 4 sample points

As we understand this comment, the purpose of your proposed added figure is to demonstrate that the 4 sample points enough to sufficiently represent the inflow and calculate the average inflow speed to the rotor. We have added a subfigure that shows the average rotor inflow speed of a downstream turbine as is moves across the wake of a downstream turbine. We have done this for 3 number of points (1, 4, and 100), demonstrating that 4 sample points is graphically similar to 100 sample points.

Figure 8; "offset" has not been properly introduced yet

An introduction for "offset" was added before this figure.

Title section 2.4: change "Intensity" into "Intensities" (in line with Fig. 1)

This change was made in the updated manuscript.

Line 205: change "mean" into "undisturbed mean"

This was changed.

This was added.
* * *
Eq. (23): add, for clarity: TI(r)=TI_a + Delta_TI, with Delta_TI given by Eq. (11)

This was added in the explanation of the referenced equation.
* * *
Title 2.5: adopt; see 1st bullet point Major Remarks

This change was implemented, in the text and in Fig. 1.
* * *
Eq. (24): change U into U(r); + mention the equation for U(r) (is it Eq. (10)?)

Great suggestion, this was added.
* * *
Section 2.6; show in an appendix a few examples of wind speed, rotational speed, pitch angle and bending moments varying over the azimuth angle.

This would be another interesting figure! However, as was mentioned in our response to a comment before, we feel like it is not and exact fit for the this paper. Although it would be interesting, in this paper we only consider the extreme azimuth angles that are most affected by partial waking, and explain this in the text. A figure showing the full variation of these values over the azimuth, although interesting, is not necessary for this paper.
* * *
Eq. (26): adopt; see 1st bullet point Major Remarks; make 2 versions (Eq. 26a and 26b): one with TI averaged over blade (from Eq. (23)) and one with TI averaged over rotor

This change was made, and an additional equation was added to section 2.9 with the turbulent wind speeds calculated for the blade.
* * *
Line 282: I guess it should be step 4 (instead of 3)

Yes! This was corrected.
* * *
Section 2.9 / line 296: refer to Eq. (26a)

We think you mean to refer to equation 26, as there is no 26a, which we think is a good idea. This was added.
* * *
Section 2.7: refer to Eq. (26b)

We aren't sure what you mean with this comment, as there is no equation 26b, and section 2.7 is where equation 26 is defined. We haven't made any edits based on this comment.
* * *
Line 361: change step 7 into step 8

This was fixed in the revised manuscript.
* * *
Line 404 (end): typo "us"

This was fixed in the revised manuscript.
* * *
Line 442: add a statement about tower shadow

The following was added to this section:

"We assumed that tower shadow is negligible, meaning that the power is only a function of inflow wind speed with no adjustment required."
* * *
Line 495: skip "about 6 times more" (such a comparison doesn't make sense)

This was removed in the revised manuscript.
* * *
Caption Figure 17: add for clarity that 0.04 corresponds with 0.07 normalized

This was added in the revised manuscript.
* * *
Line 521: skip digit: about 5%

This was changed in the revised manuscript.
* * *
Section 5: you may add for further research: to performan several SOWFA simulation in order to determine the spread of the SOWFA results (Fig. 6, 9, 10 and 11)

Good suggestion. The following was added to the proposed future work paragraph:

"First, further validate and improve our proposed damage model with more SOWFA runs for a wide variety of wind conditions. In this paper we have presented a range of wind speeds, amounts partial waking, distances downstream, and two ambient turbulence intensities. Further confidence could be achieved with more high fidelity data."
* * *
Line 631: is wind shear included?

Yes, the shear exponent we used was 0.12. We added this to the appendix.

**Reviewer 2**

First, we would like the express our sincere thanks for reviewing our paper. We know it is a time consuming process, and we are extremely grateful for your thorough and thoughtful review. We have structured this response to be clear and easy to follow. Each of your original comments will be shown in red, immediately followed by our response in black.

**General comments:**

The authors present an interesting approach to make efficient fatigue estimation for wind farms viable; in particular for use in optimization-based design approaches or possibly in future systems for operation and maintenance. By utilizing analytical models and empirical surrogate models, the study provides a simplified, but transparent methodology for estimating the blade fatigue and shows how this can be used in a simplified optimization context to make decisions about the wind farm layout. A few of the subsections could use additional details and explanations and, as will be explained below, there is a common theme of error estimation that could be considered and/or underlined to strengthen the paper.

**Specific comments:**

What exactly is shown in Fig 2? The y-axis label is not clear and hence it is not clear what is even measured on the y-axis. The caption indicates that this is an "example set" of turbulence samples, but the text on the same page indicates that the figure shows exactly the samples used in the paper. Which is it?

We agree, this was unclear. We added to the caption of Figure 2 to clarify this. The new caption reads:

"The set of turbulence samples, $S$, used in this study. Turbulence intensity is defined as TI $= \sigma_u/\bar{u}$, where $\sigma_u$ is the standard deviation in wind speeds over a given time, and $\bar{u}$ is the mean wind speed. These turbulence samples are used in a future step to calculate an instantaneous wind speed adjusted for turbulence as $u_i = U_{\text{steady}}(1 + S_i \text{ TI})$."

Any further comments on the Gaussian Wake model (Page 8-11)? It is indicated that the authors "found good results" with it, presumably when comparing with the SOWFA data (?). Are there any effects that this model is expected to miss? Do you have error estimates for the tuning constants in Tab 1? It would be instructive to include these in the table (as +/-) or show the overall effect on the surrogate fit by having error bars in Fig 4 and 5. If the errors are very small, a short comment to this effect in the text would suffice.

Excellent, we agree this could use some more clarification/justification. We have done the following to address these comments:

- After initially introducing the wake model, we added: "Overall, this model performs very well at capturing the velocity profile in the wake of a turbine, matching high fidelity data very well. For our purposes, the most important physical effects that this model does not capture is inflow flow heterogeneity, which can affect power production and loads."

- We added a small mention of the comparison of the fit for the different turbulence cases that we considered.

- We added the $R^2$ value of the curve fit to Table 1, indicating how well each model profile fits the SOWFA data.

Page 11, eq 10 and below: The authors provide a reference to justify the use of the linear wake summation model, but a few more comments here would be instructive (whether the content of these can be found in the reference or not). E.g. what is the motivation for using the linear model over others besides the fact that it "works well with the Gaussian wake model"? Is it used for superior accuracy alone or is it more a case of a simpler model that works acceptably well without introducing further complications? Is there any downside to using this model?

The following text was added to clarify this point:

"This wake combination method has been shown to compare well with experimental data when combined with the Gaussian wake model we used (citation). Additionally, this combination method is equally computationally efficient wake combination methods, such as taking the two-norm of the wake deficits."

Similarly as above, any further comments on the turbulence intensity model (Page 12-14) chosen and any possible impact of this choice? Any error estimates for the tuning constants in Tab 3 (or possibly the overall effect on the error level of the model shown in Fig 9)?

The results are fairly sensitive to the turbulence, as you might expect in a scenario of partial waking. In our damage calculations, this means that yes, the results are sensitive to the turbulence model. This is hard to do, as turbulence data is typically quite noisy. With the model, we simply want to get as close as we can to reality with a simple analytic expression. To show the accuracy of the turbulence model that we used, we included the $R^2$ values for each of the fits and for each distance downstream of the waking turbine. A really interesting area for further work would be to explore the sensitivity of the damage with respect to ambient turbulence intensity and wake turbulence behavior.

What is the expected error level of the surrogate model described in Section 2.9 on Page 17?

We have added the $R^2$ value of each fit to the caption of the figure visualizing the surrogate to give an idea of the error.

It might be instructive to illustrate a bit more clearly how a load/moment "history" is obtained via the Turbulence and Azimuth Loop, perhaps through some example. Specifically, how this method produces something analogous to the conventional load time series obtained from simulations that are usually the input to rainflow counting-based fatigue assessment methods.

The following has been added to the introduction of the loop calculating the blade loads (1.6) to orient the reader and explain this:

"The steps in this loop are to: 1.7) calculate the turbine inflow wind speed accounting for turbulence, 1.8) using this inflow speed, determine the turbine rotational speed and blade pitch, and 1.9) determine average turbulent wind speed across a blade, and use this speed and the blade pitch in the loads surrogate to determine the blade loads at the time step. These steps are then repeated for as many azimuth angles and rotations that will be simulated. After each time through the loop, the loads calculated in step 1.9 are added to a loads history. The end result is a history of the flatwise and edgewise blade loads, which is used in future steps to make fatigue calculations. "

It is indicated on Page 19 (line 360) that the results in the paper are based on load histories obtained from 50 complete revolutions of the rotor. For the NREL 5MW this would be something like a few minutes of simulation time. Conventional time domain-based fatigue estimates are usually based on at least 60 minutes

of simulation. The shorter duration is understandable for the purposes of the paper, but do the authors have any comment on this?

Great point, we agree that this is important to acknowledge in the paper. The following was added on this point:

"Although conventional time domain-based fatigue estimates are generally based on a longer time period of simulation, for the purposes of this paper in which we demonstrate the use of this proposed model, this shorter time was used for decreased computational expense."
* * *
Fig 10 and 11: The y-axis labels should indicate that the values shown are in fact lifetime fatigue values (which is my assumption, but this is not clear).

That is correct. This point was clarified in the caption of the figures.
* * *
In the Conclusion, on Page 30, the use of active yaw control and its possible coupling to the proposed method is discussed for future research. As noted previously in the paper, the yaw angle was fixed at zero in this study. Any comment on what effect (if any) non-zero yaw angles (or yaw errors even) might have on the proposed method?

Great question. We believe it would be a fairly straightforward addition to add in considerations for non-zero yaw angles or yaw errors. From the loads model, the only things that would be required would be to create a 2-D surrogate of the loads, making the blade loads a function of the inflow wind speed and the yaw. The wake deflection that would occur would be accounted for in the analytic wake model, and we believe every other portion of the model could remain as presented.
* * *
I have made several comments concerning error estimates for various parts of the proposed method. Beyond the general interest of such error estimates as indicators for the validity of each simplification, the analytical nature of the authors' methodology actually makes it possible to potentially propagate these errors all the way to the end fatigue result. The resulting error estimates could be very useful and would in fact be a strength of the method. In particular, for optimization it could provide some manner of error bound or expected uncertainty in the result that would show the level of robustness of the solution. Especially in light of Fig 14 and related results. It could also make optimization approaches that consider uncertainty more explicitly more viable for use in similar wind farm studies. While any larger investigation into this issue or indeed carrying out such an error propagation might be out of the scope of the present work, some discussion of these points would be favorable for the paper.

This is a great comment! We agree, considering error estimates and uncertainty is an is important for a model such as this, which is sensitive to the model parameters and uncertainty in inputs. Although, as you state, any significant exploration of this topic is beyond the scope of this paper, we have added to the future work paragraph in the conclusions the following to at least address this topic:

"Fifth, investigate the sensitivities and uncertainties involved with each of the models and assumptions made throughout the model, and how they impact the final damage calculations. This would be incredibly relevant for future studies that specifically include uncertainty analysis. The method presented in this paper uses analytic models, but we expect that the final results are sensitive to model parameters, tuning variables, and uncertainty in any inputs. A better understanding of these uncertainties would be important in building reliable wind farms."
* * *
Do the authors have any comments regarding the use of SOWFA as a benchmark for the accuracy of the proposed method and to what extent SOWFA itself can be used in this manner (i.e. its accuracy)? It is likely worth pointing out that any relevant experimental data for windfarms, which may not be available at

present, could presumably be similarly used to tune the parameters of the method, so it is not reliant on SOWFA as such.

Thanks for pointing this out, we did not provide sufficient justification for using SOWFA as our validation. In the revised manuscript, we added a sentence on how SOWFA has been previously validated along with citations. Additionally, we added to the future work section a portion about comparing our model to real wind farms, rather than exclusively to SOWFA.

---

## Referee Report (RR1)

**Review of the paper entitled: "A Model to Calculate Fatigue Damage Caused by Partial Waking during Wind Farm Optimization"**

**General comment:**

The paper can be divided into two parts. The first deals with the development of a simplified model to evaluate the fatigue loads of turbines in wind farms, with special emphasis on wake-induced fatigue. In the second part, the simplified model is used to perform farm design optimizations constrained by fatigue.

The idea behind this paper is well worth considering and the manuscript can be certainly cited in future publications in this field.

While the second part is well-done and easy to follow, the first part, related to the development of the simplified fatigue model, deserves some modifications. I indicated some "Important comments" mainly connected to the first part of the paper.

I also had the opportunity to check the previous review run along with the modifications implemented by the Authors to accommodate Reviewers' comments. In this regard, I would like to stress that an indication of mine (see in my review "Important comment #3") refers to a previous reviewer's comment that, in my opinion, was not adequately addressed during the first review round.

I recommend acceptance with minor revisions, but I strongly hope the Authors will pay great attention to "Important comment #3".

**Important comments:**

1. Figure 5 and related analyses: The performed analysis is correct to evaluate the goodness of the wake model. I was wondering whether it is possible to use directly the profiles generated by LES in a look-up table fashion. In fact, from Tab. 2, it seems that the model is tuned separately at each speed and each turbulence intensity. The estimated parameters do not show a clear behavior with respect to the speed, even in the low TI case, which is the one associated to the best agreement. This is an index of the poorness of the tuning process. Sentence of lines 241-243 ("the damage model is … while demonstrating our damage model.") offers another link to what I suggested at the beginning: why not using directly the profile extracted from LES?

2. Figure 7 represents an analysis of paramount importance for the paper, as it provides the justification for the use of a low-order representation of the wake shape. By the way, it is hard to understand what is meant with "final damage values", especially because the damage is addressed in a subsequent part of the paper (see Sections 2.10, 2.11, 2.12). A better explanation is needed. Moreover, why is the comparison made only between 4 and 300 samples. A fair comparison should have been done between these two cases and the SOWFA data (considered as the ground truth).

3. Line 320 to end of Section 2.6: in my opinion this period is not entirely free of errors and may be prone to misinterpretations. I will list my doubts in the following.
   a. "… two azimuth angles of 90 degrees and 270 degrees are sufficient to predict the fatigue damage". This sentence comes "out-of-the-blue", first because there is no demonstration for that throughout the paper, and second because, at least at a first sight, it seems a simplistic view. I may frankly say that one is not able to capture a single load cycle with only two samples. To have a complete picture one would need a third piece of information (cf. the Coleman transformation). In any case, even considering three or four samples, the picture results incomplete as well since the higher frequency content may have a significant impact on fatigue.

   i. Suggestion: since azimuth 90 and 270 degrees are strictly connected to fatigue induced by partial wake impingement, maybe the Authors can smooth a bit the sentence and refer explicitly to the impact of wakes and not the fatigue in general. This is actually the focus of the paper, and stressing this here may be fair, otherwise one may erroneously think that the simplified model can be used for accurately predicting the entire fatigue of a turbine for rotor design activity. I guess that this is not the Authors' intention.

   ii. Suggestion: for the problem at hand, it is important to capture the trend of fatigue with respect to the impingement level, rather than the "real" fatigue. The authors may play a bit around this concept to stress the adequacy of the approach.

   iii. Suggestion: is it simple to extend the methodology including more angles? If so, this should be reported. Moreover, what is the expected penalization in computational time induced by the inclusion of more azimuthal samples?

  b. "… at this angle [0 and 180] the moments due to gravity are zero". This can be true for in-plane loads. For out-of-plane loads, if a turbine has precone and/or tilt, the gravitational loads are maximum exactly at 0 and 180 deg. Due to dynamics of the rotor there is also a small delay in the response of the blades. Finally, since the blade may pitch, out-of-plane and in-plane loads mix together into blade flap- and edge-wise. Authors' sentence is a good approximation for edgewise, low pitch angles and low frequency vibrations.

  c. "However, for most conditions, just considering these two azimuth angles is sufficient as they capture the largest load differences which contribute the most to fatigue damage". Here again as in point a, it is hard to demonstrate that with the analyses hitherto explained. If the Authors refer to the sole impact of wake and for the goal of the work (not for characterizing the whole "fatigue damage"), then the sentence can be acceptable. But in the present form, the text should be amended.

4. Table 2 and 3: There are negative values in the $R^2$ metrics. For decent models, usually, $R^2$ is within 0 and 1, where 1 is a perfect description of data and 0 refers to a model correctness as good as the data mean value. Negative $R^2$ values are associated to extremely poor predictions. A better comment should be added to explain the obtained $R^2$ metrics.

**Minor comments:**

1. Line 58: "Current fatigue load prediction models are computationally expensive and not suitable for use in an optimization framework". The Authors may be interested in publication https://doi.org/10.5194/wes-4-549-2019, where it is presented a fast evaluation of fatigue based on pre-computed look-up table. Clearly, define the complete set of LUTs with the different types of wakes and overlapping levels is time consuming, but, once computed and stored, using them in an optimization is straightforward.

2. The Authors used "Flapwise" (1 time) and "Flatwise" (10 times) throughout the paper. Is this correct? Could it be uniformized?

3. Figure 6: a reader should benefit from the knowledge of the location of the single and 100 points.

4. Figure 7: missing x-label.

5. Figure 9 and 10: unit of measure in the y-axis?

6. Figure 9 and 10: there is a consistent overestimation at about +0.5 D. In the text, the Authors mention that this can be due to gravity. Isn't it possible that shear layer be responsible for that?

7. Figure 9 and 10: it is not clear whether the plots refer to flap- or edge-wise loads.

8. Problem 38: last constraint: it is not clear whether the Authors are constraining the edge- or flap-wise moment.

---

## Author Response (AR2)

**Response to Reviewers**

Andrew P. J. Stanley, Jennifer King, Christopher Bay, and Andrew Ning

January 2022

    Again we would like the express our gratitude for the reviews of our paper. We realize that many people took time out of a busy schedule to read this manuscript and provide feedback, for which we are very grateful. As with our original response to the previous round of comments, we have structured this response to be clear and easy to follow. Each of the original comments will be shown in blue, immediately followed by our response in black.

**Reviewer 1**

No reviewer comments

**Reviewer 2**

Some of the $R^2$ values that were added in the revised manuscript are negative and in one case the negative value is less than -1. Usually $R^2$ values are in (0,1), but values such as those reported can occur e.g. when fitting non-linear models. The authors note that some of these fits are not as good as others, but make no explicit mention of negative $R^2$. A small comment noting/explaining these entries in Tables 2 and 3 would be helpful for any reader who might be confused at the presence of negative $R^2$ values.

In the last manuscript, we had incorrectly calculated the $R^2$ values separately for each of the separation distances, when only one set tuning parameters was calculated for the wind speed. This has been corrected, and the tables now report one $R^2$ value for each wind speed instead of 3. All $R^2$ values in the updated manuscript are between 0–1.

**Reviewer 3**

**Important Comments**

1. Figure 5 and related analyses: The performed analysis is correct to evaluate the goodness of the wake model. I was wondering whether it is possible to use directly the profiles generated by LES in a look-up table fashion. In fact, from Tab. 2, it seems that the model is tuned separately at each speed and each turbulence intensity. The estimated parameters do not show a clear behavior with respect to the speed, even in the low TI case, which is the one associated to the best agreement. This is an index of the poorness of the tuning process. Sentence of lines 241-243 ("the damage model is ... while demonstrating our damage model.") offers another link to what I suggested at the beginning: why not using directly the profile extracted from LES?

We have added the following text to the paper to respond to this question:

"Because we have compared all of our intermediate models, as well as the final damage calculations, to the high-fidelity SOWFA and OpenFAST data, one might wonder why we did not directly use some surrogate of the SOWFA and OpenFAST data instead of the lower-fidelity intermediate models, or even create a surrogate directly of the final fatigue damage. These possible methods would likely provide accurate results,

and a surrogate would be computationally efficient for use during an optimization. However, a primary purpose of our model is to provide a method to estimate fatigue damage while leaving open the possibility of using computationally efficient analytic models. Our model does not require the user to run computationally expensive, complex, and high-fidelity simulations, although they certainly could. With our method, simple analytic models can be used to sufficiently estimate fatigue damage from partial waking, given that the intermediate analytic models are sufficiently accurate. For this paper, we use tuning constants to improve the comparison of our analytic models to our SOWFA data, which did require us to generate the high-fidelity data. However, for many or most applications where this fatigue model, such model tuning and exact match of previously generated data would be unnecessary."
* * *
2. Figure 7 represents an analysis of paramount importance for the paper, as it provides the justification for the use of a low-order representation of the wake shape. By the way, it is hard to understand what is meant with "final damage values", especially because the damage is addressed in a subsequent part of the paper (see Sections 2.10, 2.11, 2.12). A better explanation is needed. Moreover, why is the comparison made only between 4 and 300 samples. A fair comparison should have been done between these two cases and the SOWFA data (considered as the ground truth).

Great comment and questions. The following sentence has been added to address the issue of "final damage values" being presented at this point of the paper before the rest of the fatigue model has been described:

"The damage values shown in this figure are calculated with the model fully presented in the rest of this section. Even though the full details of this model are presented in the subsections below, we determined that it is appropriate to present this information here to demonstrate the minimal effect that the number of wind speed samples across the swept rotor area has on the final result."

As for the rest of this comment, Figure 7 is meant to demonstrate the effect of the number of wind speed samples used to calculate the effective wind speed across the rotor has on the final damage value, which we still believe it accomplishes. What are now Figures 10 and 11 provide what is suggested in the rest of your comment. These figures show the damage predicted by our reduced order models compared to the "ground truth," which we assume is sufficiently represented by the SOWFA data.
* * *
3. Line 320 to end of Section 2.6: in my opinion this period is not entirely free of errors and may be prone to misinterpretations. I will list my doubts in the following.

a. ". . . two azimuth angles of 90 degrees and 270 degrees are sufficient to predict the fatigue damage". This sentence comes "out-of-the-blue", first because there is no demonstration for that throughout the paper, and second because, at least at a first sight, it seems a simplistic view. I may frankly say that one is not able to capture a single load cycle with only two samples. To have a complete picture one would need a third piece of information (cf. the Coleman transformation). In any case, even considering three or four samples, the picture results incomplete as well since the higher frequency content may have a significant impact on fatigue.

i. Suggestion: since azimuth 90 and 270 degrees are strictly connected to fatigue induced by partial wake impingement, maybe the Authors can smooth a bit the sentence and refer explicitly to the impact of wakes and not the fatigue in general. This is actually the focus of the paper, and stressing this here may be fair, otherwise one may erroneously think that the simplified model can be used for accurately predicting the entire fatigue of a turbine for rotor design activity. I guess that this is not the Authors' intention.

ii. Suggestion: for the problem at hand, it is important to capture the trend of fatigue with respect to the impingement level, rather than the "real" fatigue. The authors may play a bit around this concept to stress the adequacy of the approach.

iii. Suggestion: is it simple to extend the methodology including more angles? If so, this should be reported. Moreover, what is the expected penalization in computational time induced by the inclusion of more azimuthal samples?

b. "... at this angle [0 and 180] the moments due to gravity are zero". This can be true for inplane loads. For out-of-plane loads, if a turbine has precone and/or tilt, the gravitational loads are maximum exactly at 0 and 180 deg. Due to dynamics of the rotor there is also a small delay in the response of the blades. Finally, since the blade may pitch, out-of-plane and in-plane loads mix together into blade flap- and edge-wise. Authors' sentence is a good approximation for edgewise, low pitch angles and low frequency vibrations.

c. "However, for most conditions, just considering these two azimuth angles is sufficient as they capture the largest load differences which contribute the most to fatigue damage". Here again as in point a, it is hard to demonstrate that with the analyses hitherto explained. If the Authors refer to the sole impact of wake and for the goal of the work (not for characterizing the whole "fatigue damage"), then the sentence can be acceptable. But in the present form, the text should be amended.

Thanks for this comment and the associated suggestions. We completely agree, it is important that in the paper we discuss and justify our decision to evaluate two azimuth angles during during a full rotor rotation. We have added a figure (what is now Figure 9) and the necessary associated text discussing this figure which explains and justifies our decision to use two azimuth angles, the possibility of evaluating with more azimuth angles, and the computational expense. While we have not followed all of these suggestions exactly, we believe we have captured the most important and relevant parts of this comment, and believe the paper is much improved from the changes.
* * *
4. Table 2 and 3: There are negative values in the $R^2$ metrics. For decent models, usually, $R^2$ is within 0 and 1, where 1 is a perfect description of data and 0 refers to a model correctness as good as the data mean value. Negative $R^2$ values are associated to extremely poor predictions. A better comment should be added to explain the obtained $R^2$ metrics

Refer back to our response from the comment from Reviewer 2. This has been corrected, and the tables now report one $R^2$ value for each wind speed instead of 3. All $R^2$ values are between 0–1.

**Minor Comments**

1. Line 58: "Current fatigue load prediction models are computationally expensive and not suitable for use in an optimization framework". The Authors may be interested in publication https://doi.org/10.5194/wes-4-549-2019, where it is presented a fast evaluation of fatigue based on pre-computed look-up table. Clearly, define the complete set of LUTs with the different types of wakes and overlapping levels is time consuming, but, once computed and stored, using them in an optimization is straightforward.

Great addition. We have mentioned and cited this paper in the new manuscript.
* * *
2. The Authors used "Flapwise" (1 time) and "Flatwise" (10 times) throughout the paper. Is this correct? Could it be uniformized?

Yes that is correct, the use of "flapwise" refers to the moments returned by CCBlade.
* * *
3. Figure 6: a reader should benefit from the knowledge of the location of the single and 100 points.

We added to this figure showing the location of the single and 100 points, in addition to the 4 point locations.

**4. Figure 7: missing x-label.**

The missing x-ticks were added to the figure.

**5. Figure 9 and 10: unit of measure in the y-axis?**

We added labels to the y-axis indicating that the figure shows damage values.

**6. Figure 9 and 10: there is a consistent overestimation at about +0.5 D. In the text, the Authors mention that this can be due to gravity. Isn't it possible that shear layer be responsible for that?**

There was a numerical error in the previous version of the velocity tuning. After correction, the damage values (especially for the low turbulence intensity examples) from our model match the high-fidelity SOWFA data much better. Even so, we added the following sentence to clarify that wind shear could account for some of the differences between our model and the high-fidelity data:

"Another possible reason for the difference between our model and the SOWFA data is wind shear, which is better captured with the high-fidelity SOWFA simulation."

**7. Figure 9 and 10: it is not clear whether the plots refer to flap- or edge-wise loads.**

These figures show the total damage calculated by our model, which takes into account both flatwise and edgewise loads. The following sentence was added to clarify this:

"Remember that the damages shown in these figures is the total damage calculated by our model, which takes into account both the flatwise and edgewise loads."

**8. Problem 38: last constraint: it is not clear whether the Authors are constraining the edge- or flapwise moment.**

It is the total damage which takes into account both the flatwise and edgewise moments. The following sentence was added to clarify this:

"This fatigue damage was calculated with our model presented in this paper, which takes into account both the flatwise and edgewise loads."